# Improved Bounds on Neural Complexity for Representing Piecewise Linear Functions

**Kuan-Lin Chen**
Department of Electrical and Computer Engineering
University of California, San Diego
La Jolla, CA 92093, USA
kuc029@ucsd.edu

**Harinath Garudadri**
Qualcomm Institute
University of California, San Diego
La Jolla, CA 92093, USA
hgarudadri@ucsd.edu

**Bhaskar D. Rao**
Department of Electrical and Computer Engineering
University of California, San Diego
La Jolla, CA 92093, USA
brao@ucsd.edu

## Abstract

A deep neural network using rectified linear units represents a continuous piecewise linear (CPWL) function and vice versa. Recent results in the literature estimated that the number of neurons needed to exactly represent any CPWL function grows exponentially with the number of pieces or exponentially in terms of the factorial of the number of distinct linear components. Moreover, such growth is amplified linearly with the input dimension. These existing results seem to indicate that the cost of representing a CPWL function is expensive. In this paper, we propose much tighter bounds and establish a polynomial time algorithm to find a network satisfying these bounds for any given CPWL function. We prove that the number of hidden neurons required to exactly represent any CPWL function is at most a quadratic function of the number of pieces. In contrast to all previous results, this upper bound is invariant to the input dimension. Besides the number of pieces, we also study the number of distinct linear components in CPWL functions. When such a number is also given, we prove that the quadratic complexity turns into bilinear, which implies a lower neural complexity because the number of distinct linear components is always not greater than the minimum number of pieces in a CPWL function. When the number of pieces is unknown, we prove that, in terms of the number of distinct linear components, the neural complexities of any CPWL function are at most polynomial growth for low-dimensional inputs and factorial growth for the worst-case scenario, which are significantly better than existing results in the literature.

## 1 Introduction

The rectified linear unit (ReLU) [Fukushima, 1980, Nair and Hinton, 2010] activation has been by far the most widely used nonlinearity and successful building block in deep neural networks (DNNs). Numerous architectures based on ReLU DNNs have achieved remarkable performance or state-of-the-art accuracy in speech processing [Zeiler et al., 2013, Maas et al., 2013], computer vision [Krizhevsky et al., 2012, Simonyan and Zisserman, 2015, He et al., 2016], medical image segmentation [Ronneberger et al., 2015], game playing [Mnih et al., 2015, Silver et al., 2016], and natural language processing [Vaswani et al., 2017], just to name a few. Besides such unprecedented

36th Conference on Neural Information Processing Systems (NeurIPS 2022).

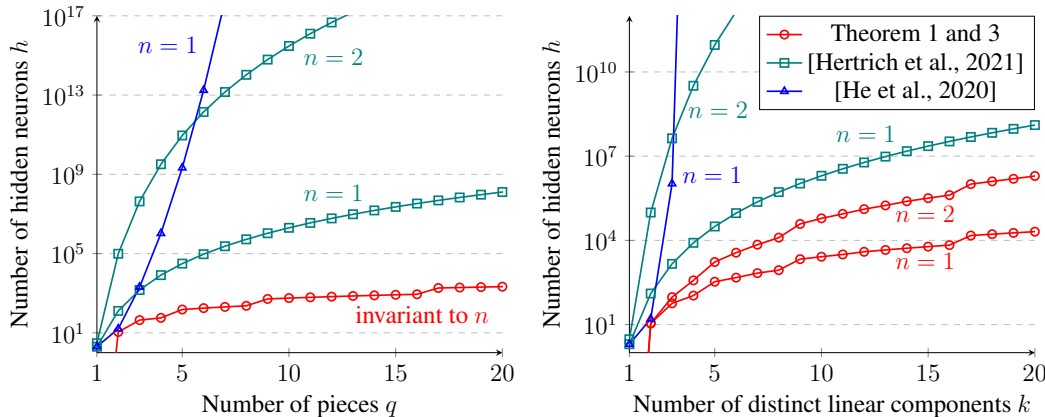

Figure 1: Any CPWL function $\mathbb{R}^n \to \mathbb{R}$ with $q$ pieces or $k$ distinct linear components can be exactly represented by a ReLU network with at most $h$ hidden neurons. In Theorem 1 and 3, $h = 0$ when $q = 1$ or $k = 1$. The bounds in Theorem 1 and the worst-case bounds in Theorem 3 are invariant to $n$. (4) is used to infer $h$ based on the depth and width given by Hertrich et al. [2021]. The upper bounds given by Theorem 1 and 3 are substantially lower than existing bounds in the literature, implying that any CPWL function can be exactly realized by a ReLU network at a much lower cost.

empirical success, ReLU DNNs are also probably the most understandable nonlinear deep learning models due to their ability to be "un-rectified" [Hwang and Heinecke, 2019].

The ability to demystify ReLU DNNs via "un-rectifying ReLUs" dates back to a seminal work by Pascanu et al. in 2014. Because each of ReLUs in a hidden layer divides the space of the preceding layer's output into two half spaces whose ReLU response is affine in one half space and exactly zero in the other, the layer of ReLUs can be replaced by an input-dependent diagonal matrix whose diagonal elements are ones for firing ReLUs and zeros for non-firing ReLUs. Based on this rationale, Pascanu et al. [2014] proved that a neural network using ReLUs divides the input space into many linear regions such that the network itself is an affine function within every region. Two excellent visualizations are shown in Figure 2 in [Hanin and Rolnick, 2019a] and [Hanin and Rolnick, 2019b]. At this point, it is quite evident that any ReLU network exactly represents a CPWL function. Pascanu et al. also proved that the maximum number of linear regions for any ReLU network with a single hidden layer is equivalent to the number of connected components induced by arrangements of hyperplanes in general position where each hyperplane corresponds to a ReLU in the hidden layer. Such a number can be computed in a closed form by Zaslavsky's Theorem [Zaslavsky, 1975]. Furthermore, they showed that the maximum number of linear regions can be bounded from below by exponential growth in terms of the number of hidden layers, leading to a conclusion that ReLU DNNs can generate more linear regions than their shallow counterparts. In the same year, Montúfar et al. improved such a lower bound and gave the first upper bound for the maximum number of linear regions. These bounds and their assumptions were later improved by [Montúfar, 2017, Raghu et al., 2017, Arora et al., 2018, Serra et al., 2018, Hinz and van de Geer, 2019], just to name a few. We refer readers to Hinz's doctoral thesis for a thorough discussion on the upper bound of the number of linear regions. Because a CPWL function with more pieces can better approximate any given continuous function and a ReLU DNN exactly represents a CPWL function [Arora et al., 2018], a ReLU DNN with more linear regions in general exhibits stronger expressivity. In summary, this "un-rectifying" perspective provides us a new angle to understand ReLU DNNs, and the results in some ways align with advances in approximation theory demonstrating the expressivity.[1]

Despite these advancements in linear regions, the complexity of a ReLU DNN that exactly represents a given CPWL function remains largely unexplored. One can find that this question is the opposite direction of the above-mentioned line of research. Although Arora et al. [2018] proved that any CPWL function can be exactly represented by a ReLU DNN with a bounded depth, any estimates

---

[1]The approximation viewpoint is not the focus of this paper. The literature on approximation is vast and we refer readers to [Vardi et al., 2021, Lu et al., 2017, Eldan and Shamir, 2016, Telgarsky, 2016, Hornik et al., 1989, Cybenko, 1989], just to name a few.

regarding the width or number of neurons of such a network were not given. The resources required for a ReLU neural network to exactly represent a CPWL function remained unknown until He et al. [2020] provided a bound to the complexity of a ReLU network that realizes any given CPWL function. They proved that the number of neurons is bounded from above by exponential growth in terms of the product between the number of pieces and the number of distinct linear components of a given CPWL function. Such an exponential bound also grows linearly with the input dimension. Because the number of pieces is an upper bound of the number of distinct linear components for any CPWL function [Tarela and Martínez, 1999, He et al., 2020], the bound grows exponentially with the quadratic number of pieces, which seems to imply that the cost for representing a CPWL function by a ReLU DNN is exceedingly high.

The most recent upper bound can be inferred from a recent work by Hertrich et al. [2021] although the number of hidden neurons was not directly given. Hertrich et al. [2021] proved a width bound in terms of the number of distinct linear components under the same depth used by Arora et al. [2018] and He et al. [2020].[2] In particular, they proved that the maximum width of a ReLU network that represents any given CPWL function can be polynomially bounded from above in terms of the number of distinct linear components. However, the order of such a polynomial is a quadratic function of the input dimension, which can be immensely large for a small number of pieces or linear components when the input dimension is large. This bound grows larger with the input dimension even though the underlying CPWL function is just a one-hidden-layer ReLU network using only one ReLU (see Figure 1 for the difference between $n = 1$ and $n = 2$ when $q = 2$ or $k = 2$).

In this paper, we provide improved bounds showing that any CPWL function can be represented by a ReLU DNN whose neural complexity is bounded from above by functions with much slower growth (see Figure 1). Our results imply that one can exactly realize any given CPWL function by a ReLU network at a much lower cost. On the other hand, in addition to guaranteeing the existence of such a network, we also give a *polynomial time* algorithm to exactly find a network satisfying our bounds. To the best of our knowledge, our results regarding the computational resource for a ReLU network, i.e., the number of hidden neurons, are the lowest upper bounds in the existing literature and the algorithm is the first tailored procedure to find a network representation from any given CPWL function. Key results and main contributions of this paper are highlighted below.

## 1.1 Key results and contributions

**Quadratic bounds.** We prove that any CPWL function with $q$ pieces can be represented by a ReLU network whose number of hidden neurons is bounded from above by a quadratic function of $q$. We also give the corresponding upper bounds for the maximum width, i.e., the maximum number of neurons per hidden layer, and the number of layers for such a network. The maximum width is bounded from above by $\mathcal{O}(q^2)$ and the number of layers is bounded from above by a logarithmic function of $q$, i.e., $\mathcal{O}(\log_2 q)$. *These bounds are invariant to the input dimension.* For any affine function, the upper bounds for the maximum width and the number of hidden neurons are zero.

**Further improvements on neural complexity.** When the number of distinct linear components $k$ of any CPWL function is given along with the number of pieces $q$, the quadratic bounds $\mathcal{O}(q^2)$ for the number of hidden neurons and the maximum width turn into *bilinear* bounds of $k$ and $q$, i.e., $\mathcal{O}(kq)$. Such a change reduces the neural complexity because $k \leq q$, and $q$ can be much larger than $k$. Still, these bounds are independent of the input dimension.

**Finding a network satisfying bilinear bounds.** We establish a polynomial time algorithm that finds a ReLU network representing any given CPWL function. The network found by the algorithm satisfies the bilinear bounds on the number of hidden neurons and the maximum width, and the logarithmic bound on the number of layers. Note that such an algorithm also guarantees that one can always reverse-engineer at least one ReLU network from the function it computes. Compared to the general-purpose reverse-engineering algorithm proposed by Rolnick and Kording [2020], our algorithm specializes in the situation when pieces of a CPWL function are given.

---

[2]The number of "affine pieces" used by Theorem 4.4 in [Hertrich et al., 2021] should be interpreted as the number of distinct linear components to best reflect the upper bound for the maximum width. Such an interpretation of "affine pieces" is different from the convention used by Pascanu et al. [2014], Montúfar et al. [2014], Arora et al. [2018], Hanin and Rolnick [2019a], and this work.

**Improved bounds from a perspective of linear components.** When the number of pieces of a CPWL function is unknown and only the number of linear components $k$ is available, we prove that the number of hidden neurons and maximum width are bounded from above by factorial growth. More precisely, $\mathcal{O}\left(k \cdot k!\right)$. The number of layers is bounded from above by linearithmic growth, or $\mathcal{O}(k \log_2 k)$. However, when the input dimension $n$ grows sufficiently slower than $k$, e.g., $\mathcal{O}\left(\sqrt{k}\right)$, then bounds for the number of hidden neurons and maximum width reduce to *polynomial* growth functions of order $2n + 1$; and the linearithmic growth reduces to $\mathcal{O}\left(n \log_2 k\right)$ for the depth.

**A new approach to choosing the depth.** Instead of scaling the depth of a ReLU network with the input dimension [Arora et al., 2018, He et al., 2020, Hertrich et al., 2021], we reveal that constructing a ReLU network whose depth is scaled with the number of pieces of the given CPWL function is more advantageous. Such a scaling turns out to be the key to deriving better upper bounds. This insight is provided by the max-min representation of CPWL functions [Tarela et al., 1990]. The importance of this scaling on the depth in ReLU networks has not been well recognized by existing bounds in the literature. We discuss implications of different representations in Section 4.

## 2 Preliminaries

Notation and definitions used in this paper are set up and clarified in this section. The set $\{1, 2, \cdots, m\}$ is denoted by $[m]$. $\mathbb{I}\left[condition\right]$ is an indicator function that gives 1 if the *condition* is true, and 0 otherwise. The CPWL function is defined by Definition 1 below.

**Definition 1.** *A function $p\colon \mathbb{R}^n \to \mathbb{R}$ is said to be CPWL if there exists a finite number of closed subsets of $\mathbb{R}^n$, say $\{\mathcal{U}_i\}_{i\in[m]}$, such that (a) $\mathbb{R}^n = \bigcup_{i\in[m]} \mathcal{U}_i$; (b) $p$ is affine on $\mathcal{U}_i, \forall i \in [m]$.*

A family of closed *convex* subsets, say $\{\mathcal{X}_i\}_{i\in[q]}$, satisfying Definition 1 is also referred to as a family of convex regions, affine pieces or simply *pieces* for a CPWL function in this paper. Definition 1 follows the definition of CPWL functions by Ovchinnikov [2002]. Notice that there are different definitions in the literature. For example, Chua and Deng [1988] and Arora et al. [2018] defined a CPWL function on a finite number of polyhedral regions. However, their definitions are essentially the same as Definition 1 because any family of closed subsets satisfying Definition 1 can be decomposed into polyhedral regions. It is possible that some of the closed subsets satisfying Definition 1 are non-convex even though the number of them reaches the minimum (see Figure 2 in [Wang and Sun, 2005]). The continuity is implied by Definition 1 due to the subsets being closed.

Because the goal of this paper is to bound the complexity of a ReLU DNN that exactly represents any given CPWL function, it is necessary to be able to measure the complexity of a CPWL function. The complexity of a CPWL function can be described using two different perspectives. One is the number of pieces $q$, which is the number of closed convex subsets satisfying Definition 1. Because this number has a minimum and any finite number above the minimum can be a valid $m$ in Definition 1, the bounds become obviously loose when the number of pieces is not the minimum. Without loss of generality, we are interested in the number $q$ when it is the minimum. The other is the number of distinct linear components $k$. A linear component of a CPWL function is defined in Definition 2.

**Definition 2.** *An affine function $f$ is said to be a linear component of a CPWL function $p$ if there exists a nonempty subset $\mathcal{M} \subseteq [m]$ such that $f(\mathbf{x}) = p(\mathbf{x}), \forall \mathbf{x} \in \bigcup_{i\in\mathcal{M}} \mathcal{U}_i$ where $\{\mathcal{U}_i\}_{i\in[m]}$ is a family of the minimum number of closed subsets satisfying Definition 1.*

A greater $q$ or $k$ gives a CPWL function more degrees of freedom because a CPWL function allowed to use $q + 1$ pieces or $k + 1$ arbitrary linear components can represent any CPWL function with $q$ pieces or $k$ distinct linear components and still have the flexibility to modify existing affine maps or increase the number of distinct affine maps of the CPWL function. Although increasing them both leads to a CPWL function with greater flexibility, the speed of upgrading degrees of freedom is different from each other. Note that a CPWL function with $q$ pieces can never have more than $q$ distinct linear components and a CPWL function with $k$ distinct linear components can easily have more than $k$ minimum number of pieces. Such a difference in a 1-dimensional case can be clearly observed from Figure 1 in [Tarela and Martínez, 1999]. Note that it is possible for two disjoint subsets from a minimum number of closed subsets satisfying Definition 1 to share the same linear component. In other words, a linear component can be reused by multiple pieces. Hence, increasing $k$ gives faster growth than increasing $q$ for the complexity and expressivity of CPWL functions.

We define the ReLU activation function in Definition 3. The ReLU network defined in Definition 4 is a simple architecture which is usually referred to as a *ReLU multi-layer perceptron*. Definition 5 defines the corresponding number of hidden neurons, depth, and maximum width.

**Definition 3.** *The rectified linear unit (ReLU) activation function $\sigma$ is defined as $\sigma(x) = \max(0, x)$. The ReLU layer or vector-valued rectified linear activation function $\sigma_k$ is defined as $\sigma_k(\mathbf{x}) = \begin{bmatrix} \sigma(x_1) & \sigma(x_2) & \cdots & \sigma(x_k) \end{bmatrix}^\mathsf{T}$ where $\mathbf{x} = \begin{bmatrix} x_1 & x_2 & \cdots & x_k \end{bmatrix}^\mathsf{T}$.*

**Definition 4.** *Let $l$ be any positive integer. A function $g\colon \mathbb{R}^{k_0} \to \mathbb{R}^{k_l}$ is said to be an $l$-layer ReLU network if there exist weights $\mathbf{W}_i \in \mathbb{R}^{k_i \times k_{i-1}}$ and $\mathbf{b}_i \in \mathbb{R}^{k_i}$ for $i \in [l]$ such that the input-output relationship of the network satisfies $g(\mathbf{x}) = h_l(\mathbf{x})$ where $h_1(\mathbf{x}) = \mathbf{W}_1 \mathbf{x} + \mathbf{b}_1$ and $h_i(\mathbf{x}) = \mathbf{W}_i \sigma_{k_{i-1}} \left( h_{i-1}(\mathbf{x}) \right) + \mathbf{b}_i$ for every $i \in [l] \setminus [1]$.*

**Definition 5.** *The sum $\sum_{l=1}^{L-1} k_l$ and the maximum $\max_{l \in [L-1]} k_l$ for $L > 1$ are referred to as the number of hidden neurons and the maximum width of an $L$-layer ReLU network, respectively. Any 1-layer ReLU network is said to have $0$ hidden neurons and a maximum width of $0$. An $l$-layer ReLU network is said to have depth $l$ and $l - 1$ hidden layers.*

## 3 Upper bounds on neural complexity for representing CPWL functions

The correspondence between CPWL functions and ReLU networks was first clearly confirmed by Theorem 2.1 in [Arora et al., 2018] although a weaker version of the correspondence can be inferred from Proposition 4.1 in [Goodfellow et al., 2013]. Arora et al. [2018] proved that every ReLU network $\mathbb{R}^n \to \mathbb{R}$ exactly represents a CPWL function, and the converse is also true, i.e., every CPWL function can be exactly represented by a ReLU network. One of the key steps used by Arora et al. [2018] to construct a ReLU network from any given CPWL function relies on an important representation result by Wang and Sun [2005], stating that any CPWL function can be represented by a sum of a finite number of *max-$\eta$-affine* functions [Magnani and Boyd, 2009] whose signs may be flipped and $\eta$ is bounded from above by $n + 1$ where $\eta$ is the number of affine functions in the *max-$\eta$-affine* function. The implication of using this representation is later discussed in Section 4.1 and its *max-$\eta$-affine* functions are given therein. The bound $\eta \leq n + 1$ in the representation allowed Arora et al. [2018] to further prove that there exists a ReLU DNN with at most

$$\lceil \log_2(n + 1) \rceil \tag{1}$$

hidden layers to exactly realize any given CPWL function. However, the computational resource required for a ReLU network to exactly represent any CPWL function had not been available in the literature until the work by He et al. [2020].

### 3.1 Upper bounds in prior work

He et al. [2020] proved that a CPWL function $\mathbb{R}^n \to \mathbb{R}$ with $q$ pieces and $k$ linear components can be represented by a ReLU network whose number of neurons is given by

$$\begin{cases} \mathcal{O}\left(n 2^{kq + (n+1)(k-n-1)}\right), & \text{if } k \geq n + 1, \\ \mathcal{O}\left(n 2^{kq}\right), & \text{if } k < n + 1. \end{cases} \tag{2}$$

The number of hidden layers in such a ReLU DNN is also bounded from above by $\lceil \log_2(n + 1) \rceil$, which is the same as the bound derived by Arora et al. [2018]. One of their significant contributions in our view is that they utilize the number of pieces and linear components of a CPWL function to bound the complexity of the equivalent ReLU network. He et al. [2020] also proved the relationship

$$k \leq q \leq k! \tag{3}$$

for any CPWL function. Note that the bounds in (3) on the number of pieces $q$ and linear components $k$ were first mentioned by Tarela and Martínez [1999] who developed the lattice representation of CPWL functions. Asymptotically, the bounds in (2) for $k \geq n + 1$ and $k < n + 1$ are amplified linearly with the input dimension $n$ for any fixed $k$. Due to (3), they can be further bounded from above by $\mathcal{O}\left(n 2^{q^2 + (n+1)(q-n-1)}\right)$ and $\mathcal{O}\left(n 2^{q^2}\right)$ in terms of $q$ and $n$. On the other hand, in terms of $k$ and $n$, they can be further bounded from above by $\mathcal{O}\left(n 2^{k \cdot k! + (n+1)(k-n-1)}\right)$ and $\mathcal{O}\left(n 2^{k \cdot k!}\right)$.

Because these bounds grow much faster than exponential growth, they seem to suggest that the cost of computing a CPWL function via a ReLU network is exceptionally high.

Hertrich et al. [2021] proved that any CPWL function $\mathbb{R}^n \to \mathbb{R}$ with $k$ distinct linear components can be represented by a ReLU network whose maximum width is $\mathcal{O}\left(k^{2n^2+3n+1}\right)$ under the same number of hidden layers $\lceil \log_2(n+1) \rceil$. Hence, the number of hidden neurons must be bounded from above by

$$\mathcal{O}\left(k^{2n^2+3n+1}\log_2(n+1)\right). \tag{4}$$

Note that we infer this bound by taking the product of the depth and the maximum width. Using $k \le q$, the bound in (4) can be expressed in terms of $q$, leading to $\mathcal{O}\left(q^{2n^2+3n+1}\log_2(n+1)\right)$. Such a bound can grow slower than $\mathcal{O}\left(n2^{q^2}\right)$, but it grows faster than $\mathcal{O}\left(n2^{q^2}\right)$ if the input dimension $n$ grows sufficiently faster than the number of pieces $q$. Also, $\mathcal{O}\left(k^{2n^2+3n+1}\log_2(n+1)\right)$ grows faster than $\mathcal{O}\left(n2^{k\cdot k!}\right)$ when the input dimension $n$ grows sufficiently faster than the number of distinct linear components $k$.

### 3.2 Improved upper bounds

We show that any CPWL function can be represented by a ReLU network whose number of hidden neurons is bounded by much slower growth functions. We state our main results in Theorem 1, 2 and 3, and focus on their impact in this subsection. Each one of them is tailored to a specific complexity measure of the CPWL function. Their proof sketches are deferred to Section 4.2. We first focus on the case when the number of linear components is unknown and the complexity of the CPWL is only measured by the number of pieces $q$.

**Theorem 1.** *Any CPWL function $p \colon \mathbb{R}^n \to \mathbb{R}$ with $q$ pieces can be represented by a ReLU network whose number of layers $l$, maximum width $w$, and number of hidden neurons $h$ satisfy*

$$l \le 2\lceil \log_2 q \rceil + 1, \tag{5}$$

$$w \le \mathbb{I}\left[q > 1\right]\left\lceil \frac{3q}{2} \right\rceil q, \tag{6}$$

*and*

$$h \le \left(3 \cdot 2^{\lceil \log_2 q \rceil} + 2\lceil \log_2 q \rceil - 3\right)q + 3 \cdot 2^{\lceil \log_2 q \rceil} - 2\lceil \log_2 q \rceil - 3. \tag{7}$$

*Furthermore, Algorithm 1 finds such a network in $\mathrm{poly}\,(n,q,L)$ time where $L$ is the number of bits required to represent every entry of the rational matrix $\mathbf{A}_i$ in the polyhedron representation $\{\mathbf{x} \in \mathbb{R}^n | \mathbf{A}_i\mathbf{x} \le \mathbf{b}_i\}$ of the piece $\mathcal{X}_i$ for every $i \in [q]$.*

---

**Algorithm 1** Find a ReLU network that computes a given continuous piecewise linear function

---

**Input:** A CPWL function $p$ with pieces $\{\mathcal{X}_i\}_{i \in [q]}$ of $\mathbb{R}^n$ satisfying Definition 1.
**Output:** A ReLU network $g$ computing $g(\mathbf{x}) = p(\mathbf{x}), \forall \mathbf{x} \in \mathbb{R}^n$.
  1: $f_1, f_2, \cdots, f_k \leftarrow$ run Algorithm 6 to find all distinct linear components of $p$
  2: **for** $i = 1, 2, \cdots, q$ **do**
  3:      $\mathcal{A}_i \leftarrow \emptyset$
  4:      **for** $j = 1, 2 \cdots, k$ **do**
  5:          **if** $f_j(\mathbf{x}) \ge p(\mathbf{x}), \forall \mathbf{x} \in \mathcal{X}_i$ **then**
  6:              $\mathcal{A}_i \leftarrow \mathcal{A}_i \bigcup \{j\}$
  7:          **end if**
  8:      **end for**
  9:      $v_i \leftarrow$ run Algorithm 2 with $\{f_m\}_{m \in \mathcal{A}_i}$ using the minimum type
 10: **end for**
 11: $v \leftarrow$ run Algorithm 3 with $v_1, v_2, \cdots, v_q$            $\triangleright$ Combine $q$ ReLU networks in parallel
 12: $u \leftarrow$ run Algorithm 2 with $\left\{\begin{bmatrix} s_1 & s_2 & \cdots & s_q \end{bmatrix}^\top \mapsto s_m\right\}_{m \in [q]}$ using the maximum type
 13: $g \leftarrow$ run Algorithm 4 with $v$ and $u$          $\triangleright$ Find a ReLU network for the composition $u \circ v$

---

The proof of Theorem 1 is deferred to Appendix B.4 in the supplementary material. Algorithm 6, 2, 3, and 4 used by Algorithm 1 are deferred to Appendix C in the supplementary material and will be discussed soon after the discussion on bounds. Because $2^{\lceil \log_2 q \rceil} < 2q$, the upper bound in (7) can be further bounded from above by $6q^2 + 2 \lceil \log_2 q \rceil q + 3q - 2 \lceil \log_2 q \rceil - 3$, leading to the asymptotic bound $h = \mathcal{O}(q^2)$. Obviously, $l = \mathcal{O}(\log_2 q)$ and $w = \mathcal{O}(q^2)$. Since the bound given by Theorem 5.2 in He et al. [2020] can be lower bounded by $\mathcal{O}\left(n2^{q^2}\right)$, it grows exponentially faster than our bound of $h$ given in Theorem 1. On the other hand, the upper bound given by (4) is at least polynomially larger than our bound of $h$ and the order of this polynomial grows quadratically with the input dimension $n$. Note that such a polynomial becomes an exponential function when the growth in $n$ is not slower than $q$. Such differences are illustrated by the figure on the left-hand side of Figure 1. The bounds in Theorem 1 are independent of the input dimension $n$. Hence, one can realize any given CPWL function using a relatively small ReLU network even though $n$ is huge.

In terms of the maximum width, the upper bound given by (6) is at least polynomially smaller than the one given by Hertrich et al. [2021]. In contrast to the bound for the number of layers in [Arora et al., 2018, He et al., 2020, Hertrich et al., 2021] that grows logarithmically with the input dimension $n$, our bound in Theorem 1 grows logarithmically with the number of pieces $q$. Therefore, the ReLU network found by Algorithm 1 in general becomes deeper when the CPWL becomes more complex for a fixed input dimension. On the other hand, the network remains the same depth even for an arbitrarily larger $n$ as long as $q$ is fixed. Taking an affine function for example, a 1-layer ReLU network with 0 hidden neurons is the solution given by Theorem 1. However, the bound given by [Arora et al., 2018, He et al., 2020, Hertrich et al., 2021] keeps increasing the depth for a larger $n$.

We briefly explain algorithms used by Algorithm 1. Algorithm 2 finds a ReLU network that computes a *max-affine* or *min-affine* function [Magnani and Boyd, 2009]. Algorithm 3 concatenates two given ReLU networks in parallel and returns another ReLU network computing the concatenation of two outputs. Algorithm 4 finds a ReLU network that represents a composition of two given ReLU networks. These algorithms are basic manipulations of ReLU networks. Algorithm 1 is a polynomial time algorithm, following from the proof of Theorem 2. Table 1 in Appendix C in the supplementary material gives a complexity analysis for Algorithm 1.

Notice that Algorithm 1 does not need to be given any linear components or completely know the CPWL function because every distinct linear component can be found by Algorithm 6, which only needs to be given a closed $\epsilon$-ball in the interior of every piece of a CPWL function $p$ and observe the output of $p$ when feeding an input. Algorithm 6 solves a system of linear equations for every piece of $p$ to find the corresponding linear component. Every system of linear equations here has a unique solution because the interior of each of the pieces is nonempty. The nonemptyness is guaranteed by Lemma 12(a) in Appendix A in the supplementary material.

The 5th step of Algorithm 1 can be executed by checking the optimization result of the following linear programming problem

$$
\begin{aligned}
\text{minimize} \quad & f_j(\mathbf{x}) - p(\mathbf{x}), \\
\text{subject to} \quad & \mathbf{x} \in \mathcal{X}_i.
\end{aligned}
\tag{8}
$$

The condition in the 5th step can only be true when the optimal value is nonnegative. Because every piece of $p$ is given to Algorithm 1, the piece $\mathcal{X}_i$ is available for the linear program as a system of linear inequalities. The objective function is also available since $p$ is affine on $\mathcal{X}_i$ and all distinct linear components are available from Algorithm 6. The corresponding linear component of $p$ on $\mathcal{X}_i$ can be found by first feeding at most $n + 1$ affinely independent points from the closed $\epsilon$-ball to $p$ and every candidate linear component, and then matching their output values.

The ellipsoid method [Khachiyan, 1979], the interior-point method [Karmarkar, 1984], and the path-following method [Renegar, 1988] are polynomial time algorithms for the linear programming problem using rational numbers on the Turing machine model of computation. These algorithms are also known to be *weakly polynomial time* algorithms. The strongly polynomial time algorithm requested by Smale's 9th problem [Smale, 1998], i.e., *the linear programming problem*, is still an open question. Given that we run the 5th step of Algorithm 1 by solving the linear programming problem in (8), Algorithm 1 is a weakly polynomial time algorithm. The question of whether it is a strongly polynomial time algorithm is not known. The dependency on the number of bits $L$ in the time complexity of Algorithm 1 directly comes from using (8) to execute the 5th step. In practice, linear programming problems can be solved very reliably and efficiently [Boyd and Vandenberghe,

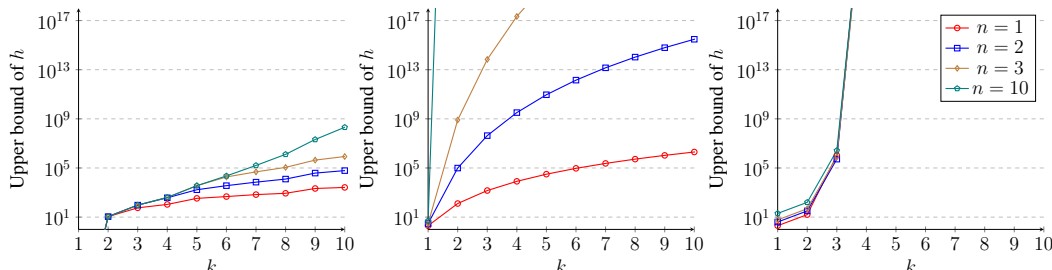

Figure 2: Left: The upper bound of $h$ in Theorem 3 grows much slower when $n$ grows sufficiently slower than $k$, leading to a much better upper bound compared to the worst-case asymptotic bound $\mathcal{O}\left(k \cdot k!\right)$ in Theorem 3 when $n$ is sufficiently larger. Middle: the bound in (4) inferred from [Hertrich et al., 2021]. Right: Theorem 5.2 in [He et al., 2020].

2004]. We provide an implementation of Algorithm 1 and measure its run time on a computer for different numbers of pieces and input dimensions in Appendix D in the supplementary material.

Theorem 2 discusses the case when the number of linear components and pieces are both known.

**Theorem 2.** *Any CPWL function* $p\colon \mathbb{R}^n \to \mathbb{R}$ *with $k$ linear components and $q$ pieces can be represented by a ReLU network whose number of layers $l$, maximum width $w$, and number of hidden neurons $h$ satisfy* $l \leq \lceil \log_2 q \rceil + \lceil \log_2 k \rceil + 1$, $w \leq \mathbb{I}\left[k > 1\right] \left\lceil \frac{3k}{2} \right\rceil q$, *and*

$$h \leq \left(3 \cdot 2^{\lceil \log_2 k \rceil} + 2 \lceil \log_2 k \rceil - 3\right) q + 3 \cdot 2^{\lceil \log_2 q \rceil} - 2 \lceil \log_2 k \rceil - 3. \tag{9}$$

*Furthermore, Algorithm 1 finds such a network in* $\mathrm{poly}\left(n, k, q, L\right)$ *time where $L$ is the number of bits required to represent every entry of the rational matrix $\mathbf{A}_i$ in the polyhedron representation* $\{\mathbf{x} \in \mathbb{R}^n | \mathbf{A}_i \mathbf{x} \leq \mathbf{b}_i\}$ *of the piece $\mathcal{X}_i$ for every $i \in [q]$.*

The proof of Theorem 2 is deferred to Appendix B.3 in the supplementary material. The bounds in Theorem 2 are in general tighter and always no worse than those in Theorem 1 because $q$ is never less than $k$ but can be much larger than $k$. Asymptotically, $l = \mathcal{O}(\log_2 q)$, $w = \mathcal{O}(kq)$, and $h = \mathcal{O}(kq)$. The bound given by Theorem 5.2 in He et al. [2020] increases exponentially faster than the bound of $h$ in Theorem 2.

When the number of linear components is the only complexity measure of the CPWL function, we resort to Theorem 3 below.

**Theorem 3.** *Any CPWL function* $p\colon \mathbb{R}^n \to \mathbb{R}$ *with $k$ linear components can be represented by a ReLU network whose number of layers $l$, maximum width $w$, and number of hidden neurons $h$ satisfy* $l \leq \lceil \log_2 \phi(n,k) \rceil + \lceil \log_2 k \rceil + 1$, $w \leq \mathbb{I}\left[k > 1\right] \left\lceil \frac{3k}{2} \right\rceil \phi(n,k)$, *and*

$$h \leq \left(3 \cdot 2^{\lceil \log_2 k \rceil} + 2 \lceil \log_2 k \rceil - 3\right) \phi(n,k) + 3 \cdot 2^{\lceil \log_2 \phi(n,k) \rceil} - 2 \lceil \log_2 k \rceil - 3 \tag{10}$$

*where*

$$\phi(n,k) = \min\left(\sum_{i=0}^{n} \binom{\frac{k^2-k}{2}}{i}, k!\right). \tag{11}$$

The proof of Theorem 3 is deferred to Appendix B.5 in the supplementary material. Because $\phi(n,k) \leq k!$, the worst-case asymptotic bounds for $l$, $w$ and $h$ are $l = \mathcal{O}\left(k \log_2 k\right)$, $w = \mathcal{O}\left(k \cdot k!\right)$, and $h = \mathcal{O}\left(k \cdot k!\right)$, respectively. However, it holds that $\sum_{i=0}^{n} \binom{\frac{k^2-k}{2}}{i} \leq k^{2n}$, so the asymptotic bounds are $l = \mathcal{O}\left(n \log_2 k\right)$, $w = \mathcal{O}\left(k^{2n+1}\right)$, and $h = \mathcal{O}\left(k^{2n+1}\right)$ when $n$ grows sufficiently slower than $k$. For example, $n = \mathcal{O}\left(\sqrt{k}\right)$. In this case, $w$ and $h$ are bounded from above by a polynomial of order $2c\sqrt{k} + 1$ for some constant $c$, which grows slower than factorial growth. Such an advantage for small $n$ is illustrated by the figure on the left-hand side of Figure 2.

Since the bound given by Theorem 5.2 in [He et al., 2020] can be bounded from below by $\mathcal{O}\left(n2^{k \cdot k!}\right)$, it at the minimum grows exponentially larger than the upper bound of $h$ in Theorem 3. Even for

a small $n$, the relative order of growth is gigantic. The figure on the right-hand side of Figure 2 illustrates such a large difference. For $k = 5$, $n2^{k \cdot k!} \approx 7.92 \times 10^{28}$ when $n = 1$, while our bound of $h$ is at most 3615 for any $n$. The difference is extremely large even though $k$ is small under $n = 1$. The middle plot in Figure 2 shows that (4) increases much faster when $n$ becomes larger. For $k = 3$, $k^{2n^2+3n+1} \log_2 (n + 1) \approx 8.20 \times 10^{110}$ when $n = 10$, while our bound of $h$ is at most 95 for any $n$. The upper bound of $h$ in Theorem 3 is much better than (4) for any $n$ and $k$.

**Lemma 1.** *Let $\mathcal{P}_{n,k}$ be the set of all CPWL functions with exactly $k$ distinct linear components such that $p \colon \mathbb{R}^n \to \mathbb{R}, \forall p \in \mathcal{P}_{n,k}$. Let $\mathcal{C}_{n,k}(p)$ be the collection of all families of closed convex subsets satisfying Definition 1 for any $p \in \mathcal{P}_{n,k}$. Then, $k \leq \min_{\mathcal{Q} \in \mathcal{C}_{n,k}(p)} |\mathcal{Q}| \leq \phi(n, k)$.*

The proof of Lemma 1 is deferred to Appendix B.1 in the supplementary material. Clearly, $\phi(n, k)$ is a better upper bound of $q$ compared to the bound $q \leq k!$ given by He et al. [2020]. When $n$ grows sufficiently slower than $k$, the bound $\phi(n, k)$ can be exponentially smaller than $k!$.

### 3.3 Limitations

Although these new bounds are significantly better than previous results, it is still possible to find a ReLU network whose hidden neurons are fewer than the bounds in Theorem 2 to exactly represent a given CPWL function. A tight bound for the case when $n = 1$ was first given by Theorem 2.2 in [Arora et al., 2018]. However, it seems more difficult to bound the size of a network from below for $n > 1$. To the best of our knowledge, we are not aware of any tight bounds in the literature for the size of the ReLU network representing a general CPWL function using an arbitrary input dimension.

## 4 Representations of CPWL functions have different implications on depth

We reveal implications of using different representations of CPWL functions and their impact on constructing ReLU networks. We first discuss the popular representation used by prior work and the implicit constraint imposed by such a representation.

### 4.1 Constrained depth

Arora et al. [2018], He et al. [2020], and Hertrich et al. [2021] proved the same bound for the number of layers, relying on the following representation of a CPWL function

$$p(\mathbf{x}) = \sum_{j=1}^{J} \sigma_j \max_{i \in \eta(j)} f_i(\mathbf{x}) \tag{12}$$

where $\sigma_j \in \{+1, -1\}$ and $\eta(j)$ is a subset of $[J]$ such that $|\eta(j)| \leq n + 1$ for all $j \in [J]$. That is, a sum of a finite number of *max-$\eta$-affine* functions whose signs may be flipped. (12) was established by Theorem 1 in [Wang and Sun, 2005] which is essentially the same as Theorem 1 in [Wang, 2004] that emphasizes the difference between two convex piecewise linear functions. This result was also used by Goodfellow et al. [2013] to prove Proposition 4.1 in the maxout network paper.

The depth given by (1) does not scale with the complexity of a CPWL function. This feature directly comes from using a ReLU network to realize each of *max-$\eta$-affine* functions in (12) and concatenating all of them together. Because the size of $\eta(j)$ is bounded from above by $n + 1$, the depth can be made to depend solely on $n$. Such a treatment seems to be the only way if one considers a CPWL function represented by (12). As a result, the ReLU network is forced to use a depth constrained by the input dimension to represent the given CPWL function, which in turn requires more hidden neurons. Because we do not use (12), our networks are not limited by such an implication.

### 4.2 Proof sketch for the unconstrained depth

We give a proof sketch in this subsection for our main results. By using a different representation, the depth of a ReLU network is able to be scaled with the complexity measure, i.e., the number of pieces, of any given CPWL function to accommodate the high expressivity.

By Theorem 4.2 in [Tarela and Martínez, 1999], any CPWL function $p$ can be represented as

$$p(\mathbf{x}) = \max_{\mathcal{X} \in \mathcal{Q}} \min_{i \in \mathcal{A}(\mathcal{X})} f_i(\mathbf{x}) \tag{13}$$

for all $\mathbf{x} \in \mathbb{R}^n$ where $\mathcal{A}(\mathcal{X}) = \{i \in [k] \mid f_i(\mathbf{x}) \geq p(\mathbf{x}), \forall \mathbf{x} \in \mathcal{X}\}$ is the set of indices of linear components that have values greater than or equal to $p(\mathbf{x})$ for all $\mathbf{x} \in \mathcal{X}$, and $\mathcal{Q}$ is any family of closed convex subsets of $\mathbb{R}^n$ satisfying Definition 1. We have used $f_1, f_2, \cdots, f_k$ to denote the $k$ distinct linear components of $p$. Notice that Theorem 4.2 in [Tarela and Martínez, 1999] was first stated by Theorem 7 in [Tarela et al., 1990]. Both are essentially the same, but Theorem 4.2 in [Tarela and Martínez, 1999] emphasizes the convexity of each of the regions in the domain. Both theorems are also fundamentally equivalent to Theorem 2.1 in [Ovchinnikov, 2002]. Notice that one of the concluding remarks in [Ovchinnikov, 2002] pointed out that the convexity of the input space is an essential assumption. The entire space $\mathbb{R}^n$ satisfies such an assumption. In addition, Ovchinnikov pointed out that the max-min representation also holds for vector-valued CPWL functions. Hence, it is possible to generalize our bounds to vector-valued CPWL functions.

Using the representation in (13) and Lemma 2 below, we are able to prove Theorem 2 by bounding the size of $\mathcal{Q}$ and $\mathcal{A}(\mathcal{X})$. Theorem 1 and 3 can be proved by applying Lemma 1 to Theorem 2. Note that the size $|\mathcal{Q}|$ in (13) is the key for the depth of a ReLU network to be able to scale with $q$.

**Lemma 2.** *Let $m$ be any positive integer. Define $l(m) = \lceil \log_2 m \rceil + 1$, $w(m) = \mathbb{I}[m > 1] \lceil \frac{3m}{2} \rceil$, and the following sequence for any positive integer $k$,*

$$r(k) = \begin{cases} 0, & \text{if } k = 1, \\ \frac{3k}{2} + r\left(\frac{k}{2}\right), & \text{if } k \text{ is even}, \\ 2 + \frac{3(k-1)}{2} + r\left(\frac{k+1}{2}\right), & \text{if } k \neq 1 \text{ and } k \text{ is odd}. \end{cases} \tag{14}$$

*Then, there exists an $l(m)$-layer ReLU network $g \colon \mathbb{R}^n \to \mathbb{R}$ with $r(m)$ hidden neurons and a maximum width of $w(m)$ such that $g$ computes the extremum of $f_1(\mathbf{x}), f_2(\mathbf{x}), \cdots, f_m(\mathbf{x})$, i.e., $g(\mathbf{x}) = \max_{i \in [m]} f_i(\mathbf{x})$ or $g(\mathbf{x}) = \min_{i \in [m]} f_i(\mathbf{x})$ for all $\mathbf{x} \in \mathbb{R}^n$ under any $m$ scalar-valued affine functions $f_1, f_2, \cdots, f_m$. Furthermore, Algorithm 2 finds such a network in $\mathrm{poly}(m, n)$ time.*

The proof of Lemma 2 is deferred to Appendix B.2 in the supplementary material. One can also view $l(m)$, $w(m)$, and $r(m)$ as upper bounds for the number of layers, maximum width, and the number of hidden neurons. Because $r(m) < 6m - 3$ by Lemma 6, the bound for the number of hidden neurons $r(m)$ is tighter than the bound $8m - 4$ given by Lemma D.3 in [Arora et al., 2018] or Lemma 5.4 in [He et al., 2020] (these two lemmas are essentially the same). The bound for the number of layers remains the same as the one given by Lemma D.3 in [Arora et al., 2018]. By combining Lemma 2 with Lemma 3, Lemma 4, and Lemma 8, we can easily perform the same job on computing the extremum of multiple scalar-valued ReLU networks as Lemma D.3 does in [Arora et al., 2018]. Lemma 6, 3, 4, and 8 are given in Appendix A in the supplementary material.

## 5 Broader impact

Our results guarantee that any CPWL function can be exactly computed by a ReLU neural network at a more manageable cost. This assurance is crucial because CPWL functions are important tools in many applications. Such an assurance also relates DNNs closer to CPWL functions and allows researchers and engineers to understand the expressivity of DNNs from a different perspective. We focus on simple ReLU networks (ReLU multi-layer perceptrons) in this paper, but it may be possible to derive bounds for other activation functions and advanced neural network architectures such as maxout networks [Goodfellow et al., 2013], residual networks [He et al., 2016], densely connected networks [Huang et al., 2017], and other nonlinear networks [Chen et al., 2021], by making some (possibly mild) assumptions. Our contributions advance the fundamental understanding of the link between ReLU networks and CPWL functions.

## Acknowledgments and disclosure of funding

We would like to thank the anonymous reviewers for their constructive comments, Tai-Hsuan Chung for answering our mathematical questions, and Christoph Hertrich for his thoughtful comments on the time complexity of Algorithm 1 and for clarifying Theorem 4.4 in [Hertrich et al., 2021]. This work was supported in part by NSF under Grant CCF-2225617, Grant CCF-2124929, and Grant IIS-1838897, in part by NIH/NIDCD under Grant R01DC015436, and in part by KIBM Innovative Research Grant Award.

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
