# A  Lemmas

**Lemma 3.** *Let $l$ be any positive integer. There exists an $l$-layer ReLU network $g$ with $2n(l-1)$ hidden neurons and a maximum width of $2n$ such that $g(\mathbf{x}) = \mathbf{x}$ for all $\mathbf{x} \in \mathbb{R}^n$. Furthermore, Algorithm 5 finds such a network in $\mathsf{poly}(n, l)$ time.*

*Proof.* Appendix B.6. □

**Definition 6.** *Let $g_{(l,n,w)}$ denote an $l$-layer ReLU network with $n$ hidden neurons and a maximum width bounded from above by $w$.*

**Lemma 4.** *There exists $g_{\left(l_1+l_2-1, n_1+n_2, \max(w_1, w_2)\right)}$ that represents any composition of $g_{(l_1, n_1, w_1)}$ and $g_{(l_2, n_2, w_2)}$. Algorithm 4 finds such a network computing the composition in $\mathsf{poly}\left(\max(w_1, w_2), \max(l_1, l_2)\right)$ time.*

*Proof.* Appendix B.7. □

**Lemma 5.** *The sequence $r(k)$ defined by (14) is a strictly increasing sequence.*

*Proof.* Appendix B.8. □

**Lemma 6.** *For any positive integer $k$, the sequence $r(k)$ defined by (14) satisfies*

$$r(k) \leq 3\left(2^{\lceil \log_2 k \rceil} - 1\right) < 6k - 3. \tag{15}$$

*Proof.* Appendix B.9 □

**Lemma 7.** *Let $m_1$ and $m_2$ be the output dimensions of $g_{(l_1, n_1, w_1)}$ and $g_{(l_2, n_2, w_2)}$, respectively. Define*

$$l = \max(l_1, l_2), \tag{16}$$
$$w = w_j + \max(w_i, 2m_i), \tag{17}$$

*and*

$$n = n_1 + n_2 + 2m_i|l_1 - l_2|, \tag{18}$$

*where $i = \arg\min_{k \in [2]} l_k$ and $j = [2] \setminus \{i\}$. Then, there exists $g_{(l,n,w)}$ such that*

$$g_{(l,n,w)}(\mathbf{x}) = \begin{bmatrix} g_{(l_1, n_1, w_1)}(\mathbf{x}) \\ g_{(l_2, n_2, w_2)}(\mathbf{x}) \end{bmatrix} \tag{19}$$

*for all $\mathbf{x} \in \mathbb{R}^n$.*

*Proof.* Appendix B.10. □

**Lemma 8.** *Let $m_1, m_2, \cdots, m_k$ be the output dimensions of $g_{(l_1, n_1, w_1)}, g_{(l_2, n_2, w_2)}, \cdots, g_{(l_k, n_k, w_k)}$, respectively. Define*

$$l = \max_{i \in [k]} l_i, \tag{20}$$

$$w = \sum_{i \in [k]} \max(w_i, 2m_i), \tag{21}$$

*and*

$$n = \sum_{i \in [k]} n_i + 2m_i(l - l_i). \tag{22}$$

*Then, there exists $g_{(l,n,w)}$ such that*

$$g_{(l,n,w)}(\mathbf{x}) = \begin{bmatrix} g_{(l_1, n_1, w_1)}(\mathbf{x}) \\ g_{(l_2, n_2, w_2)}(\mathbf{x}) \\ \vdots \\ g_{(l_k, n_k, w_k)}(\mathbf{x}) \end{bmatrix} \tag{23}$$

*for all $\mathbf{x} \in \mathbb{R}^n$. Furthermore, Algorithm 3 finds such a network in $\mathsf{poly}\left(\max_{i \in [k]} w_i, k, l\right)$ time.*

*Proof.* Appendix B.11. □

**Lemma 9.** *Let $f_1, f_2, \cdots, f_k$ be any affine functions such that $f_i \colon \mathbb{R}^n \to \mathbb{R}$ for all $i \in [k]$. Define the set of feasible ascending orders as*

$$\mathcal{S}^n_{f_1, f_2, \cdots, f_k} = \left\{ (s_1, s_2, \cdots, s_k) \in \mathfrak{S}(k) \mid f_{s_1}(\mathbf{x}) \leq f_{s_2}(\mathbf{x}) \leq \cdots \leq f_{s_k}(\mathbf{x}), \mathbf{x} \in \mathbb{R}^n \right\} \quad (24)$$

*where $\mathfrak{S}(k)$ is the collection of all permutations of the set $[k]$. It holds true that*

$$\left| \mathcal{S}^n_{f_1, f_2, \cdots, f_k} \right| \leq \min \left( \sum_{i=0}^{n} \binom{\frac{k^2-k}{2}}{i}, k! \right). \quad (25)$$

*Proof.* Appendix B.12. □

**Lemma 10.** *If Definition 1 is satisfied for a non-affine function, then every nonempty subset has a nonempty intersection with the other subset or at least one of the other subsets.*

*Proof.* Appendix B.13. □

**Assumption 1.** *The number of closed connected subsets satisfying Definition 1 is a minimum.*

The interior and frontier (boundary) of a set $\mathcal{X}$ are denoted as $\mathrm{Int}\mathcal{X}$ and $\mathrm{Fr}\mathcal{X}$, respectively.

**Lemma 11.** *Let $f_i$ denote the affine function associated with $\mathcal{X}_i$ for $i \in [I]$ where $\{\mathcal{X}_i\}_{i \in [I]}$ is a family of closed connected subsets satisfying Assumption 1. Then, for any $i \in [I], j \in [I]$ such that $i \neq j$ and $\mathcal{X}_i \bigcap \mathcal{X}_j \neq \emptyset$,*

   *(a) $f_i$ and $f_j$ are different, and $\{\mathbf{x} \in \mathbb{R}^n \mid f_i(\mathbf{x}) = f_j(\mathbf{x})\} \neq \emptyset$.*

   *(b) $\{\mathbf{x} \in \mathbb{R}^n \mid f_i(\mathbf{x}) = f_j(\mathbf{x})\}$ is an affine subspace of $\mathbb{R}^n$ with dimension $n-1$.*

   *(c) $\mathcal{X}_i \bigcap \mathcal{X}_j \subseteq \{\mathbf{x} \in \mathbb{R}^n \mid f_i(\mathbf{x}) = f_j(\mathbf{x})\}$.*

   *(d) $\mathbf{x} \notin \mathrm{Int}\mathcal{X}_i$ and $\mathbf{x} \notin \mathrm{Int}\mathcal{X}_j$ for all $\mathbf{x} \in \mathcal{X}_i \bigcap \mathcal{X}_j$.*

*Proof.* Appendix B.14, B.15, B.16, and B.17. □

**Lemma 12.** *If a family of closed connected subsets $\{\mathcal{X}_i\}_{i \in [I]}$ satisfies Assumption 1, then, for all $i \in [I]$,*

   *(a) $\mathrm{Int}\mathcal{X}_i \neq \emptyset$.*

   *(b) $Fr\mathcal{X}_i = \bigcup_{k \in [I] \setminus i} \mathcal{X}_k \bigcap \mathcal{X}_i$.*

   *(c) $\mathrm{Int}\mathcal{X}_i \bigcap \mathrm{Int}\mathcal{X}_j = \emptyset$ for all $j \in [I]$ such that $j \neq i$.*

*Proof.* Appendix B.18, B.19, and B.20. □

**Lemma 13.** *Let $\{\mathcal{X}_i\}_{i \in [m]}$ be any finite family of subsets satisfying Assumption 1. Let $\{\mathcal{H}_j\}_{j \in [k]}$ be any finite family of affine subspaces of $\mathbb{R}^n$ with dimension $n-1$. Then, for every $i \in [m]$,*

$$\mathcal{X}_i \bigcap \left( \mathbb{R}^n \setminus \bigcup_{j \in [k]} \mathcal{H}_j \right) \neq \emptyset. \quad (26)$$

*Proof.* Appendix B.21. □

**Proposition 1.** *For any family of closed connected subsets satisfying Definition 1, all subsets are the largest closed connected subsets if and only if Assumption 1 is satisfied.*

*Proof.* Appendix B.22. □

# B   Proofs

## B.1   Proof of Lemma 1

*Proof.* Let the family of closed connected subsets $\bar{\mathcal{Q}} = \{\mathcal{X}_i\}_{i\in[I]}$ satisfy Assumption 1 for any $p \in \mathcal{P}_{n,k}$. Let the $k$ distinct linear components of $p$ be $f_1, f_2, \cdots, f_k$ and $\mathcal{H}_{lm}$ be the intersection between $f_l$ and $f_m$ for $l \in [k], m \in [k], l \neq m$. Note that every $\mathcal{H}_{lm}$ is an affine subspace of $\mathbb{R}^n$ with dimension $n-1$ (a hyperplane) or an empty set. Because the linear components are distinct, it must be true that $k \leq I$ by Definition 2. If $p$ is an affine function, then it follows that $k = \min_{\bar{\mathcal{Q}}\in\mathcal{C}_{n,k}(p)}|\bar{\mathcal{Q}}| = \phi(n,k) = 1$, the claim holds. For the non-affine case, we must have $k > 1$.

Let $\mathcal{R} = \mathbb{R}^n \setminus \mathcal{H}$ where

$$\mathcal{H} = \bigcup_{k\in[m],l\in[m],k\neq l} \mathcal{H}_{kl}. \tag{27}$$

Note that $\mathcal{H} \neq \emptyset$ according to Lemma 10 and 11(c). By Lemma 12(b), the boundary or frontier of $\mathcal{X}_i$ for $i \in [I]$ is given by

$$\mathrm{Fr}\mathcal{X}_i = \bigcup_{j\in[I]\setminus i} \left(\mathcal{X}_i \bigcap \mathcal{X}_j\right). \tag{28}$$

Because every $\mathcal{X}_i \bigcap \mathcal{X}_j$ for $i \in [I], j \in [I], i \neq j$ is a subset of some $\mathcal{H}_{lm}$ for $l \in [k], m \in [k], l \neq m$ by Lemma 11(c), it follows that the boundary of $\mathcal{X}_i$, $\mathrm{Fr}\mathcal{X}_i$, satisfies

$$\mathrm{Fr}\mathcal{X}_i \subseteq \mathcal{H} \tag{29}$$

for $i \in [I]$. The interior of $\mathcal{X}_i$, $\mathrm{Int}\mathcal{X}_i$, is a nonempty subset of $\mathbb{R}^n$ according to Lemma 12(a). Furthermore, by Lemma 13,

$$\mathcal{X}_i \bigcap \mathcal{R} \neq \emptyset. \tag{30}$$

Now, define

$$\mathcal{Z}_i = (\mathrm{Int}\mathcal{X}_i) \bigcap \mathcal{R} \tag{31}$$

for $i \in [I]$. Note that $\mathcal{Z}_i = \mathcal{X}_i \bigcap \mathcal{R} \neq \emptyset$ due to (29) and (30). Let $\mathcal{A}$ be any subset of $\mathbb{R}^n$ and $\lambda(\mathcal{A})$ be the number of connected components of $\mathcal{A}$ in $\mathbb{R}^n$. It must be true that

$$1 = \lambda(\mathcal{X}_i) \leq \lambda(\mathrm{Int}\mathcal{X}_i) \leq \lambda(\mathcal{Z}_i). \tag{32}$$

By Lemma 12(c), $\mathrm{Int}\mathcal{X}_i \bigcap \mathrm{Int}\mathcal{X}_j = \emptyset$ for $i \in [I], j \in [I], i \neq j$. We have

$$I \leq \lambda\left(\bigcup_{i\in[I]} \mathrm{Int}\mathcal{X}_i\right) = \sum_{i\in[I]} \lambda(\mathrm{Int}\mathcal{X}_i) \leq \sum_{i\in[I]} \lambda(\mathcal{Z}_i) = \lambda\left(\bigcup_{i\in[I]} \mathcal{Z}_i\right) = \lambda\left(\bigcup_{i\in[I]} \mathcal{X}_i \bigcap \mathcal{R}\right). \tag{33}$$

Notice that

$$\bigcup_{i\in[I]} \mathcal{X}_i \bigcap \mathcal{R} = \mathbb{R}^n \bigcap \mathcal{R} = \mathcal{R} \tag{34}$$

by the property $\bigcup_{i\in[I]} \mathcal{X}_i = \mathbb{R}^n$ in Definition 1. Plugging (34) into (33) leads to

$$I \leq \lambda(\mathcal{R}) \tag{35}$$

which states that $I$ is bounded from above by the number of connected components of $\mathcal{R}$ in $\mathbb{R}^n$. Notice that every component is an open convex set because every component is the intersection of a finite number of open half spaces. Therefore,

$$I = |\bar{\mathcal{Q}}| = \min_{\mathcal{Q}'\in\mathcal{C}'_{n,k}(p)}|\mathcal{Q}'| \leq \min_{\mathcal{Q}\in\mathcal{C}_{n,k}(p)}|\mathcal{Q}| \leq \lambda(\mathcal{R}) \tag{36}$$

where $\mathcal{C}'_{n,k}(p)$ denotes the collection of all families of closed connected subsets satisfying Definition 1 for any $p \in \mathcal{P}_{n,k}$. Because the ascending order of these $k$ linear components does not change within a connected component of $\mathcal{R}$, $\lambda(\mathcal{R})$ can be bounded from above by the number of feasible ascending orders. Let $\mathfrak{S}(k)$ be the collection of all permutations of the set $[k]$. It follows that

$$\lambda(\mathcal{R}) \leq \left|\left\{(s_1, s_2, \cdots, s_k) \in \mathfrak{S}(k) \,|\, f_{s_1}(\mathbf{x}) \leq f_{s_2}(\mathbf{x}) \leq \cdots \leq f_{s_k}(\mathbf{x}), \mathbf{x} \in \mathbb{R}^n\right\}\right|. \tag{37}$$

Finally, Lemma 9 proves the statement by bounding the number of feasible ascending orders. $\square$

## B.2 Proof of Lemma 2

*Proof.* It suffices to show that

$$g(\mathbf{x}) = \max_{i \in [k]} x_i. \tag{38}$$

for all $\mathbf{x} = \begin{bmatrix} x_1 & x_2 & \cdots & x_k \end{bmatrix}^\mathsf{T} \in \mathbb{R}^k$ since the composition of affine functions is still affine. The affine functions can be absorbed into the first layer of the ReLU network $g$. We prove the case for taking the maximum of $m$ real numbers since the same procedure below can be applied to prove the case of taking the minimum due to the following identity

$$\min_{i \in [k]} f_i(\mathbf{x}) = -\max_{i \in [k]} -f_i(\mathbf{x}). \tag{39}$$

Because $\max(x_1, x_2) = \max(0, x_2 - x_1) + \max(0, x_1) - \max(0, -x_1)$ for any $x_1 \in \mathbb{R}$ and $x_2 \in \mathbb{R}$, it holds true that

$$\max_{j \in [k]} x_j = \begin{cases} \max_{j \in \left[\frac{k}{2}\right]} \max_{i \in \{2j-1, 2j\}} x_i, & \text{if } k \text{ is even} \\ \max_{j \in \left[\frac{k+1}{2}\right]} \alpha(j; x_1, x_2, \cdots, x_k), & \text{if } k \text{ is odd} \end{cases} \tag{40}$$

for $x_j \in \mathbb{R}, j \in [k]$ where

$$\alpha(j; x_1, x_2, \cdots, x_k) = \begin{cases} \max_{i \in \{2j-1, 2j\}} x_i, & \text{if } j \in \left[\frac{k-1}{2}\right] \\ \max(0, x_k) - \max(0, -x_k), & \text{if } j = \frac{k+1}{2} \end{cases}. \tag{41}$$

Let $r(k)$ be the number of operations of taking the maximum between a zero and a real number, i.e., $\max(0, x), x \in \mathbb{R}$ for computing the maximum of $k$ real numbers using (40). One can find $r(2) = 3$ and $r(3) = 8$ by expanding all operations in (40). Because we do not need any maximum operations to compute the maximum over a singleton, we define $r(1) = 0$. For any positive integer $k$ such that $k \geq 2$, we have the recursion

$$r(k) = \begin{cases} \frac{3k}{2} + r\left(\frac{k}{2}\right), & \text{if } k \text{ is even} \\ 2 + \frac{3(k-1)}{2} + r\left(\frac{k+1}{2}\right), & \text{if } k \text{ is odd} \end{cases} \tag{42}$$

according to (40). Note that $r(n)$ is the number of ReLUs in a ReLU network $g$ that computes the maximum of $n$ real numbers or a *max-affine* function. The number of ReLUs here is equivalent to the number of hidden neurons according to Definition 4. We shall note that the number of ReLU layers is equivalent to the number of hidden layers.

Obviously, we only need a 1-layer ReLU network with no ReLUs to compute the maximum of a singleton. Suppose that we aim to compute the maximum of $m = 2^n$ real numbers for any positive integer $n$. Then, every time the recursion goes to the next level in (42), the number of variables considered for computing the maximum is halved. Hence, the number of ReLU layers is $n$. When $m$ is not a power of two, i.e., $2^n < m < 2^{n+1}$, then we can always construct a ReLU network with $n + 1$ ReLU layers and $2^{n+1}$ input neurons, and set weights connected to the $2^{n+1} - m$ "phantom input neurons" to zeros. Because $\lceil \log_2 m \rceil = n + 1$ for $2^n < m < 2^{n+1}$, the number of ReLU layers is $\lceil \log_2 m \rceil$ for any positive integer $m$. By Definition 5, we have $l(m) = \lceil \log_2 m \rceil + 1$.

By Lemma 5, $r(k)$ is a strictly increasing sequence. Therefore, the maximum width of the network is given by the width of the first hidden layer. When $L = 1$ or $m = 1$, the width is 0 due to Definition 5. When $L > 1$ or $m > 1$,

$$\max_{l \in [L-1]} k_l = \begin{cases} \frac{3m}{2}, & \text{if } m \text{ is even} \\ 2 + \frac{3(m-1)}{2}, & \text{if } m \text{ is odd} \end{cases}$$
$$= \left\lceil \frac{3m}{2} \right\rceil. \tag{43}$$

Algorithm 2 directly follows from the above construction. Its complexity analysis is deferred to Table 2 in Appendix C. □

### B.3 Proof of Theorem 2

*Proof.* Let $f_1, f_2, \cdots, f_k$ be $k$ distinct linear components of $p$ and $\mathcal{Q}$ be any family of closed convex subsets of $\mathbb{R}^n$ satisfying Definition 1. By Theorem 4.2 in [Tarela and Martínez, 1999], $p$ can be represented as

$$p(\mathbf{x}) = \max_{\mathcal{X} \in \mathcal{Q}} \min_{i \in \mathcal{A}(\mathcal{X})} f_i(\mathbf{x}) \tag{44}$$

for all $\mathbf{x} \in \mathbb{R}^n$ where

$$\mathcal{A}(\mathcal{X}) = \left\{ i \in [k] \mid f_i(\mathbf{x}) \geq p(\mathbf{x}), \forall \mathbf{x} \in \mathcal{X} \right\} \tag{45}$$

is the set of indices of linear components that have values greater than or equal to $p(\mathbf{x})$ for all $\mathbf{x} \in \mathcal{X}$. A thorough discussion of the representation (44) is given in Section 4.2.

According to (44), there are $|\mathcal{Q}|$ minima required to be computed where each of them is a minimum of $|\mathcal{A}(\mathcal{X})|$ real numbers. Then, the value of $p$ can be computed by taking the maximum of the resulting $|\mathcal{Q}|$ minima. We will show that these operations are realizable by a ReLU network. By Lemma 2, an $l(m)$-layer ReLU network with $r(m)$ hidden neurons and a maximum width of $w(m)$ can compute the extremum of $m$ real numbers given by $m$ affine functions.

We realize (44) in three steps. First, we create $|\mathcal{Q}|$ ReLU networks where each of them is an $l\left(|\mathcal{A}(\mathcal{X})|\right)$-layer ReLU network with $r\left(|\mathcal{A}(\mathcal{X})|\right)$ hidden neurons and a maximum width of $w\left(|\mathcal{A}(\mathcal{X})|\right)$ that computes $\min_{i \in \mathcal{A}(\mathcal{X})} f_i(\mathbf{x})$ for $\mathcal{X} \in \mathcal{Q}$. Second, we parallelly concatenate these $|\mathcal{Q}|$ networks, i.e., put them in parallel and let them share the same input to obtain a ReLU network that takes $\mathbf{x}$ and outputs $|\mathcal{Q}|$ real numbers. Finally, we create an $l\left(|\mathcal{Q}|\right)$-layer ReLU network with $r\left(|\mathcal{Q}|\right)$ hidden neurons and a maximum width of $w\left(|\mathcal{Q}|\right)$ that takes the maximum of $|\mathcal{Q}|$ real numbers.

The parallel combination of $|\mathcal{Q}|$ networks in the second step can be realized by Lemma 8. The third step can be fulfilled by Lemma 4. With the above construction, we can now count the number of layers, the upper bound for the maximum width, and the number of hidden neurons for a ReLU network that realizes $p$. The number of layers is given by

$$l\left(|\mathcal{Q}|\right) + \max_{\mathcal{X} \in \mathcal{Q}} l\left(|\mathcal{A}(\mathcal{X})|\right) - 1. \tag{46}$$

The maximum width is bounded from above by

$$\max\left( \sum_{\mathcal{X} \in \mathcal{Q}} \max\left( w\left(|\mathcal{A}(\mathcal{X})|\right), 2\right), w\left(|\mathcal{Q}|\right) \right). \tag{47}$$

The number of hidden neurons is given by

$$r\left(|\mathcal{Q}|\right) + \sum_{\mathcal{X} \in \mathcal{Q}} r\left(|\mathcal{A}(\mathcal{X})|\right) + 2\left( \max_{\mathcal{Y} \in \mathcal{Q}} l\left(|\mathcal{A}(\mathcal{Y})|\right) - l\left(|\mathcal{A}(\mathcal{X})|\right) \right). \tag{48}$$

Because $\mathcal{A}(\mathcal{X})$ for every $\mathcal{X} \in \mathcal{Q}$ is a subset of $[k]$, it holds that

$$1 \leq |\mathcal{A}(\mathcal{X})| \leq k \tag{49}$$

for all $\mathcal{X} \in \mathcal{Q}$. Therefore, the number of layers in (46) can be bounded from above by

$$l\left(|\mathcal{Q}|\right) + l(k) - 1 = \lceil \log_2 |\mathcal{Q}| \rceil + \lceil \log_2 k \rceil + 1 \tag{50}$$

where we have used the definition of the function $l$ in Lemma 2. Again, using (49), the upper bound for the maximum width in (47) can be further bounded from above by

$$\max\left( \sum_{\mathcal{X} \in \mathcal{Q}} \max\left( w(k), 2\right), w\left(|\mathcal{Q}|\right) \right) = \max\left( |\mathcal{Q}| \max\left( w(k), 2\right), w\left(|\mathcal{Q}|\right) \right)$$

$$\leq \max\left( |\mathcal{Q}| \max\left( \left\lceil \frac{3k}{2} \right\rceil, 2\right), \left\lceil \frac{3|\mathcal{Q}|}{2} \right\rceil \right) \tag{51}$$

$$= \left\lceil \frac{3k}{2} \right\rceil |\mathcal{Q}|$$

where we have used the definition of the function $w$ in Lemma 2. Note that the maximum width is zero when the number of layers is one. Finally, again, using (49), the number of neurons in (48) can be bounded from above by

$$
\begin{aligned}
r\left(|\mathcal{Q}|\right) &- 2l(k) + 2l(1) + \sum_{\mathcal{X} \in \mathcal{Q}} \left(r\left(k\right) + 2l\left(k\right) - 2l(1)\right) \\
&= r\left(|\mathcal{Q}|\right) - 2l(k) + 2l(1) + |\mathcal{Q}|\left(r\left(k\right) + 2l\left(k\right) - 2l(1)\right) \\
&= r\left(|\mathcal{Q}|\right) - 2\left\lceil\log_2 k\right\rceil + |\mathcal{Q}|\left(r\left(k\right) + 2\left\lceil\log_2 k\right\rceil\right) \\
&\leq 3\left(2^{\left\lceil\log_2|\mathcal{Q}|\right\rceil} - 1\right) - 2\left\lceil\log_2 k\right\rceil + |\mathcal{Q}|\left(3\left(2^{\left\lceil\log_2 k\right\rceil} - 1\right) + 2\left\lceil\log_2 k\right\rceil\right) \\
&= 3\left(2^{\left\lceil\log_2|\mathcal{Q}|\right\rceil} - 1\right) + 3|\mathcal{Q}|\left(2^{\left\lceil\log_2 k\right\rceil} - 1\right) + 2\left(|\mathcal{Q}| - 1\right)\left\lceil\log_2 k\right\rceil
\end{aligned}
\tag{52}
$$

where we have used Lemma 6 for the upper bound in the fourth line of (52). Expanding and rearranging terms in (52) lead to (9).

Algorithm 1 directly follows from the above construction. Its complexity analysis is deferred to Table 1 in Appendix C. $\qquad\square$

### B.4 Proof of Theorem 1

*Proof.* By Lemma 1, the number of distinct linear components $k$ is bounded from above by the number of pieces, i.e., $k \leq q$, implying that the bounds in Theorem 2 can be written in terms of $q$. Substituting $k$ with $q$ in Theorem 2 proves the claim.

According to Theorem 2, the time complexity of Algorithm 1 is $\text{poly}(n, k, q, L)$. Using the bound $k \leq q$ proves the claim for the time complexity. $\qquad\square$

### B.5 Proof of Theorem 3

*Proof.* By Lemma 1, the minimum number of closed convex subsets $q$ of a CPWL function $p\colon \mathbb{R}^n \to \mathbb{R}$ can be bounded from above by $\phi(n, k)$, i.e.,

$$
q \leq \phi(n, k) = \min\left(\sum_{i=0}^{n}\binom{\frac{k^2-k}{2}}{i}, k!\right).
\tag{53}
$$

Substituting $q$ with $\phi(n, k)$ in Theorem 2 proves the claim. $\qquad\square$

### B.6 Proof of Lemma 3

*Proof.* Obviously, a one-layer ReLU network is an affine function whose weights can be set to fulfill the identity mapping in $\mathbb{R}^n$. We prove the case when the number of layers is more than one in the next paragraph. We start with a scalar case, and then work on the vector case.

For any $x \in \mathbb{R}$, it holds that $\max(0, x) - \max(0, -x) = x$. In other words, a hidden layer of two ReLUs with $+1$ and $-1$ weights can represent an identity mapping for any scalar. For any vector input in $\mathbb{R}^n$, we can concatenate such structures of two ReLUs in parallel because the identity mapping can be decomposed into $n$ individual identity mappings from $n$ coordinates. Therefore, a two-layer ReLU network with $2n$ hidden neurons can realize the identity mapping in $\mathbb{R}^n$. Stacking such a hidden layer any number of times gives a deeper network that is still an identity mapping. Algorithm 5 follows from the above construction. Its complexity analysis is deferred to Table 5 in Appendix C. $\qquad\square$

### B.7 Proof of Lemma 4

*Proof.* Because a composition of two affine mappings is still affine, the first layer of either one of the two networks can be absorbed into the last layer of the other one if their dimensions are compatible. The resulting new network still satisfies Definition 4. The number of layers of the new network is $l_1 + l_2 - 1$. The number of hidden neurons of the new network is $n_1 + n_2$. The maximum width of the new network is at most $\max(w_1, w_2)$. Algorithm 4 follows from the above construction. Its complexity analysis is deferred to Table 4 in Appendix C. $\qquad\square$

### B.8 Proof of Lemma 5

*Proof.* For any positive even integer $k \geq 4$, it holds true that

$$r(k) - r(k-1) = \frac{3k}{2} + r\left(\frac{k}{2}\right) - 2 - \frac{3(k-2)}{2} - r\left(\frac{k}{2}\right) = 1. \tag{54}$$

For any positive odd integer $k$ such that $k \geq 3$, we have

$$r(k) - r(k-1) = 2 + \frac{3(k-1)}{2} + r\left(\frac{k+1}{2}\right) - \frac{3(k-1)}{2} - r\left(\frac{k-1}{2}\right)$$

$$= \begin{cases} 3, & \text{if } \frac{k+1}{2} \text{ is even} \\ 2 + r\left(\frac{k+1}{2}\right) - r\left(\frac{k+1}{2} - 1\right), & \text{otherwise} \end{cases} \tag{55}$$

which is strictly greater than zero. Note that (55) is greater than 0 because the equality in (55) can be applied over and over again to reach (54) or the base case $r(2) - r(1) = 3$. □

### B.9 Proof of Lemma 6

*Proof.* By Lemma 5, $r(k)$ is a strictly increasing sequence. Then, it must be true that

$$r(k) = r\left(2^{\log_2 k}\right) \leq r\left(2^{\lceil \log_2 k \rceil}\right). \tag{56}$$

According to the recursion (14), it holds that

$$\begin{aligned} r\left(2^{\lceil \log_2 k \rceil}\right) &= \frac{3}{2} \sum_{i=1}^{\lceil \log_2 k \rceil} 2^i \\ &= \frac{3}{2}\left(2^{\lceil \log_2 k \rceil + 1} - 2\right) \\ &= 3\left(2^{\lceil \log_2 k \rceil} - 1\right) \\ &< 3\left(2^{(\log_2 k) + 1} - 1\right) \\ &= 3\left(2k - 1\right). \end{aligned} \tag{57}$$

□

### B.10 Proof of Lemma 7

*Proof.* Two ReLU networks can be combined in parallel such that the new network shares the same input and the two output vectors from the two ReLU networks are concatenated together. To see this, we show that the weights of the new network can be found by the following operations. Let $\mathbf{W}_i^1$ and $\mathbf{b}_i^1$ be the weights of the $i$-th layer in $g_{(l_1, n_1, w_1)}$, and $\mathbf{W}_i^2$ and $\mathbf{b}_i^2$ are the weights of the $i$-th layer in $g_{(l_2, n_2, w_2)}$. Let $\mathbf{W}_i$ and $\mathbf{b}_i$ be the weights of the new network. Now, we find the weights for the new network. In the first layer, we construct

$$\mathbf{W}_1 = \begin{bmatrix} \mathbf{W}_1^1 \\ \mathbf{W}_1^2 \end{bmatrix} \tag{58}$$

and

$$\mathbf{b}_1 = \begin{bmatrix} \mathbf{b}_1^1 \\ \mathbf{b}_1^2 \end{bmatrix}. \tag{59}$$

For the $i$-th layer such that $1 < i \leq \min(l_1, l_2)$, we use

$$\mathbf{W}_i = \begin{bmatrix} \mathbf{W}_i^1 & \mathbf{0} \\ \mathbf{0} & \mathbf{W}_i^2 \end{bmatrix} \tag{60}$$

and

$$\mathbf{b}_i = \begin{bmatrix} \mathbf{b}_i^1 \\ \mathbf{b}_i^2 \end{bmatrix}. \tag{61}$$

If $l_1 = l_2$, then the claim is proved. If $l_1 \neq l_2$, then we stack a network that implements the identity mapping to the shallower network such that the numbers of layers of the two networks are the same. Because the network $g_{(l_i, n_i, w_i)}$ is shallower than the other network, we append $|l_1 - l_2|$ hidden layers to $g_{(l_i, n_i, w_i)}$ such that the procedure in (60) and (61) can be used. By Lemma 3, there exists an $(|l_1 - l_2| + 1)$-layer ReLU network $g_{(|l_1 - l_2| + 1, 2m_i |l_1 - l_2|, 2m_i)}$ with $2m_i |l_1 - l_2|$ hidden neurons and a maximum width bounded from above by $2m_i$ for representing the identity mapping in $\mathbb{R}^{m_i}$. By Lemma 4, there exists a network $g_{(l_i + |l_1 - l_2|, n_i + 2m_i |l_1 - l_2|, \max(w_i, 2m_i))}$ that represents the composition of $g_{(|l_1 - l_2| + 1, 2m_i |l_1 - l_2|, 2m_i)}$ and $g_{(l_i, n_i, w_i)}$. Now, (60) and (61) can be used to combine $g_{(l_j, n_j, w_j)}$ and $g_{(l_i + |l_1 - l_2|, n_i + 2m_i |l_1 - l_2|, \max(w_i, 2m_i))}$ in parallel because the number of layers in network $g_{(l_i + |l_1 - l_2|, n_i + 2m_i |l_1 - l_2|, \max(w_i, 2m_i))}$ is equal to $l_j$ according to the fact that $l_i + |l_1 - l_2| = \max(l_1, l_2) = l_j$. Such a new network has $\max(l_1, l_2)$ layers and

$$n_j + n_i + 2m_i |l_1 - l_2| = n_1 + n_2 + 2m_i |l_1 - l_2| \tag{62}$$

hidden neurons. The maximum width of the new network is at most $w_j + \max(w_i, 2m_i)$. $\qquad \square$

## B.11 Proof of Lemma 8

*Proof.* The case $k = 1$ is trivial. The case $k = 2$ is proved by Lemma 7, which gives a tighter bound on the maximum width. The number of layers and hidden neurons of the claim agree with Lemma 7 when $k = 2$. The claim can be proved by following a similar procedure from the proof of Lemma 7. By Lemma 3, we can stack an identity mapping realized by an $(l - l_i + 1)$-layer ReLU network with $2m_i(l - l_i)$ hidden neurons and a maximum width of $2m_i$ on the $i$-th network for all $i \in [k]$ such that $l_i < l$. In other words, we increase the number of hidden layers for any network whose number of layers is less than $l$ such that the cascade of the network and the corresponding identity mapping has $l$ layers. For all $i \in [k]$ such that $l_i < l$, the extended network has $n_i + 2m_i(l - l_i)$ hidden neurons and a maximum width at most $\max(w_i, 2m_i)$ according to Lemma 4. Because all the networks now have the same number of layers, we can directly combine them in parallel. Hence, the resulting new network has $\max_{i \in [k]} l_i$ layers and

$$\sum_{i \in [k]} n_i + 2m_i(l - l_i) \tag{63}$$

hidden neurons and a maximum width at most

$$\sum_{i \in [k]} \max(w_i, 2m_i). \tag{64}$$

Algorithm 3 directly follows from the above construction. Its complexity analysis is deferred to Table 3 in Appendix C. $\qquad \square$

## B.12 Proof of Lemma 9

*Proof.* Because $\mathcal{S}^n_{f_1, f_2, \cdots, f_k}$ is a subset of $\mathfrak{S}(k)$ and $\left| \mathfrak{S}(k) \right| = k!$ is the number of permutations of $k$ distinct objects, it follows that

$$\left| \mathcal{S}^n_{f_1, f_2, \cdots, f_k} \right| \leq k!. \tag{65}$$

On the other hand, the number of hyperplanes, or affine subspaces of $\mathbb{R}^n$ with dimension $n - 1$, induced by the distinct intersections between any two different affine functions is bounded from above by

$$\binom{k}{2}. \tag{66}$$

Let the arrangement of these hyperplanes be $\mathcal{A}$, and $|\mathcal{A}|$ be the number of hyperplanes in the arrangement. By Zaslavsky's Theorem [Zaslavsky, 1975], the number of connected components of the set

$$\mathbb{R}^n \setminus \bigcup_{H \in \mathcal{A}} H \tag{67}$$

is bounded from above by

$$\sum_{i=0}^n \binom{|\mathcal{A}|}{i} \tag{68}$$

Because there are at most $\binom{k}{2}$ hyperplanes in $\mathbb{R}^n$, it follows that

$$\left| \mathcal{S}^n_{f_1, f_2, \cdots, f_k} \right| \leq \sum_{i=0}^{n} \binom{\binom{k}{2}}{i}. \tag{69}$$

Combining (65) and (69) proves the claim. Notice that the ascending order does not change within a connected component. □

## B.13 Proof of Lemma 10

*Proof.* Let $\mathcal{X}_1, \mathcal{X}_2, \cdots, \mathcal{X}_I$ be a family of nonempty subsets satisfying Definition 1 for a non-affine function. We prove the claim by contradiction. Suppose that there exists at least one nonempty closed subset, say $\mathcal{X}_i$, that is disjoint with every other closed subset $\mathcal{X}_j, j \in [I] \setminus i$. It follows that

$$\mathcal{X}_i \bigcap \bigcup_{j \in [I] \setminus i} \mathcal{X}_j = \emptyset \tag{70}$$

which implies

$$\left( \mathbb{R}^n \setminus \mathcal{X}_i \right) \bigcup \left( \mathbb{R}^n \setminus \bigcup_{j \in [I] \setminus i} \mathcal{X}_j \right) = \mathbb{R}^n. \tag{71}$$

Because the union of any finite collection of closed sets is closed, it must be true that $\bigcup_{j \in [I] \setminus i} \mathcal{X}_j$ is closed. Notice that $\mathcal{X}_i$ is never the whole space $\mathbb{R}^n$ because the CPWL function is assumed to be non-affine. $\bigcup_{j \in [I] \setminus i} \mathcal{X}_j$ must be nonempty due to Definition 1. Therefore, both $\mathbb{R}^n \setminus \mathcal{X}_i$ and $\mathbb{R}^n \setminus \bigcup_{j \in [I] \setminus i} \mathcal{X}_j$ are nonempty and open. Since $\mathbb{R}^n$ is connected, it cannot be represented as the union of two disjoint nonempty open subsets. It follows that the intersection between $\mathbb{R}^n \setminus \mathcal{X}_i$ and $\mathbb{R}^n \setminus \bigcup_{j \in [I] \setminus i} \mathcal{X}_j$ is nonempty. In other words, there exists an element of $\mathbb{R}^n$ that is not in $\mathcal{X}_i$ and $\bigcup_{j \in [I] \setminus i} \mathcal{X}_j$, contradicting Definition 1. □

## B.14 Proof of Lemma 11(a)

*Proof.* If the CPWL function is affine, then there are no intersecting closed subsets because the only closed subset satisfying Assumption 1 is $\mathbb{R}^n$. On the other hand, if the CPWL function is non-affine, then there exist at least two intersecting closed subsets according to Lemma 10. For any two intersecting closed subsets, say $\mathcal{X}_i$ and $\mathcal{X}_j$, we first show that

$$\{ \mathbf{x} \in \mathbb{R}^n \mid f_i(\mathbf{x}) = f_j(\mathbf{x}) \} \neq \emptyset \tag{72}$$

where $f_i$ and $f_j$ are the affine functions corresponding to $\mathcal{X}_i$ and $\mathcal{X}_j$. We prove this statement by contradiction. Suppose that the intersection is empty, i.e., the linear equation $\left( \mathbf{a}_i - \mathbf{a}_j \right)^T \mathbf{x} + b_i - b_j = 0$ does not have a solution where $f_i(\mathbf{x}) = \mathbf{a}_i^T \mathbf{x} + b_i$ and $f_j(\mathbf{x}) = \mathbf{a}_j^T \mathbf{x} + b_j$ for $\mathbf{a}_i, \mathbf{a}_j \in \mathbb{R}^n$ and $b_i, b_j \in \mathbb{R}$. Then, it is necessary that $\mathbf{a}_i = \mathbf{a}_j$ and $b_i \neq b_j$. In other words, the two affine functions are parallel, implying that every point in $\mathcal{X}_i \cap \mathcal{X}_j$ gives two different values, which cannot be true for a valid function.

Next, we prove that there does not exist an intersection that is $\mathbb{R}^n$ by contradiction. Let us assume that there exists at least one intersection that is $\mathbb{R}^n$ between the affine functions corresponding to two intersecting closed subsets, say $\mathcal{X}_i$ and $\mathcal{X}_j$. Then, we can always replace $\mathcal{X}_i$ and $\mathcal{X}_j$ with their union. Such a replacement still satisfies Definition 1 but reduces the number of closed (connected) subsets by at least one, contradicting the fact that the number of closed subsets is a minimum. Because the two affine functions are identical if and only if the intersection is $\mathbb{R}^n$, the two affine functions must be different. □

## B.15 Proof of Lemma 11(b)

*Proof.* The claim follows from Lemma 11(a). Because the two affine functions have a nonempty intersection, their intersection must be $\mathbb{R}^n$ or an affine subspace of $\mathbb{R}^n$ with dimension $n-1$. However, the two affine functions must be different, implying that $\mathbb{R}^n$ is never the intersection. □

## B.16   Proof of Lemma 11(c)

*Proof.* Let any given two intersecting subsets be $\mathcal{X}_i$ and $\mathcal{X}_j$. The intersection between their corresponding affine functions, say $f_i$ and $f_j$, is given by $\mathcal{H}_{ij} = \{\mathbf{x} \in \mathbb{R}^n \mid f_i(\mathbf{x}) = f_j(\mathbf{x})\}$. Suppose that there exists a point $\mathbf{a} \in \mathcal{X}_i \bigcap \mathcal{X}_j$ such that $\mathbf{a} \notin \mathcal{H}_{ij}$, then it follows that $f_i(\mathbf{a}) \neq f_j(\mathbf{a})$. Such a result cannot be true for a valid function. We conclude that $\mathcal{X}_i \bigcap \mathcal{X}_j \subseteq \mathcal{H}_{ij}$.   $\square$

## B.17   Proof of Lemma 11(d)

*Proof.* We prove the statement by contradiction. Suppose there exists a point $\mathbf{c} \in \mathbb{R}^n$ in the intersection of two intersecting closed connected subsets, say $\mathcal{X}_i$ and $\mathcal{X}_j$, such that $\mathbf{c}$ is an interior point of $\mathcal{X}_i$, then there exists an open $\epsilon$-radius ball $B(\mathbf{c}, \epsilon)$ such that $\mathbf{x} \in \mathcal{X}_i, \forall \mathbf{x} \in B(\mathbf{c}, \epsilon)$ for some $\epsilon > 0$. By Lemma 11(b), the intersection between the two affine functions corresponding to $\mathcal{X}_i$ and $\mathcal{X}_j$ must be an affine subspace of $\mathbb{R}^n$ with dimension $n - 1$. Let such an affine subspace be denoted as $\mathcal{H}_{ij}$ and its corresponding linear subspace be denoted as $\mathcal{V}(\mathcal{H}_{ij})$. Then, there exists a nonzero vector $\mathbf{d} \in \mathbb{R}^n$ such that $\alpha \mathbf{d} \perp \mathbf{v}$ for all $\mathbf{v} \in \mathcal{V}(\mathcal{H}_{ij})$ and any $\alpha \neq 0$. Therefore, it follows that $\alpha \mathbf{d} + \mathbf{a} \notin \mathcal{H}_{ij}$ for any $\mathbf{a} \in \mathcal{H}_{ij}$ and any $\alpha \neq 0$. According to Lemma 11(c), $\mathcal{X}_i \cap \mathcal{X}_j \subseteq \mathcal{H}_{ij}$, so we have $\alpha \mathbf{d} + \mathbf{c} \notin \mathcal{X}_i \cap \mathcal{X}_j$ for any $\alpha \neq 0$. When $\alpha = \frac{\epsilon}{2\|\mathbf{d}\|_2}$ or $\alpha = \frac{-\epsilon}{2\|\mathbf{d}\|_2}$, $\alpha \mathbf{d} + \mathbf{c} \in B(\mathbf{c}, \epsilon)$. However, one of them must satisfy $\alpha \mathbf{d} + \mathbf{c} \notin \mathcal{X}_i$, contradicting the existence of a point in $\mathcal{X}_i \cap \mathcal{X}_j$ that is an interior point of $\mathcal{X}_i$. The same procedure can be applied to prove that there does not exist a point in $\mathcal{X}_i \cap \mathcal{X}_j$ such that it is an interior point of $\mathcal{X}_j$. We conclude that every element in $\mathcal{X}_i \cap \mathcal{X}_j$ is not an interior point of $\mathcal{X}_i$ or $\mathcal{X}_j$.   $\square$

## B.18   Proof of Lemma 12(a)

*Proof.* The boundary or frontier of $\mathcal{X}_i$ is given by

$$
\begin{aligned}
\mathrm{Fr}\mathcal{X}_i &= \overline{\mathcal{X}_i} \bigcap \overline{\mathbb{R}^n \setminus \mathcal{X}_i} \\
&= \mathcal{X}_i \bigcap \overline{\left( \bigcup_{k \in [I]} \mathcal{X}_k \right) \setminus \mathcal{X}_i} \\
&= \mathcal{X}_i \bigcap \overline{\bigcup_{k \in [I] \setminus i} \left( \mathcal{X}_k \setminus \mathcal{X}_k \bigcap \mathcal{X}_i \right)} \\
&= \mathcal{X}_i \bigcap \bigcup_{k \in [I] \setminus i} \overline{\left( \mathcal{X}_k \setminus \mathcal{X}_k \bigcap \mathcal{X}_i \right)} \qquad (73) \\
&= \mathcal{X}_i \bigcap \bigcup_{k \in [I] \setminus i} \overline{\mathcal{X}_k} \\
&= \mathcal{X}_i \bigcap \bigcup_{k \in [I] \setminus i} \mathcal{X}_k \\
&= \bigcup_{k \in [I] \setminus i} \mathcal{X}_k \bigcap \mathcal{X}_i
\end{aligned}
$$

where $\overline{\mathcal{A}}$ denotes the closure of a subset $\mathcal{A}$. We have used Lemma 11(d) for the equality between the 4-th and 5-th line of (73). Now, we prove that the interior of $\mathcal{X}_i$ is nonemtpy by contradiction. Suppose that the interior of $\mathcal{X}_i$ is empty, then it follows that $\mathcal{X}_i = \overline{\mathcal{X}_i} = \mathrm{Fr}\mathcal{X}_i$ because the closure of $\mathcal{X}_i$ is the union of the interior and the boundary of $\mathcal{X}_i$. Combining that with (73), we have $\mathcal{X}_i = \bigcup_{k \in [I] \setminus i} \mathcal{X}_k \bigcap \mathcal{X}_i$. which implies every element in $\mathcal{X}_i$ is at least covered by one of the other closed subsets $\mathcal{X}_k$ for some $k \in [I] \setminus i$. In this case, we can delete $\mathcal{X}_i$ from $\mathcal{X}_1, \mathcal{X}_2, \cdots, \mathcal{X}_I$; and the remaining $I - 1$ closed subsets still satisfy Definition 1. Such a valid deletion of $\mathcal{X}_i$ contradicts the fact that $I$ is the minimum number of closed subsets. Hence, the interior of $\mathcal{X}_i$ must be nonempty.   $\square$

## B.19   Proof of Lemma 12(b)

*Proof.* The statement is proved by (73) in Lemma 12(a).   $\square$

### B.20 Proof of Lemma 12(c)

*Proof.* By Lemma 12(a), the interior of every subset is nonempty. Next, by Lemma 11(d), every point in the intersection between any two subsets is a boundary point of both subsets. It follows that the interiors of any two subsets are disjoint. $\square$

### B.21 Proof of Lemma 13

*Proof.* By Lemma 12(a), the interior of $\mathcal{X}_i$ is nonempty. Therefore, there exists an open $\epsilon$-radius ball $B(\mathbf{c}_0, \epsilon)$ such that $\mathbf{x} \in \mathcal{X}_i, \forall \mathbf{x} \in B(\mathbf{c}_0, \epsilon)$ for some $\epsilon > 0$ and $\mathbf{c}_0 \in \mathcal{X}_i$. Let us consider the set

$$\bigcap_{j \in [k]} \left( B(\mathbf{c}_0, \epsilon) \bigcap \left( \mathcal{H}_j^+ \bigcup \mathcal{H}_j^- \right) \right) \tag{74}$$

where $\mathcal{H}_j^+$ and $\mathcal{H}_j^-$ are two open half spaces created by $\mathcal{H}_j$. It suffices to show the nonemptyness of the set in (74) to prove the claim. If $\mathcal{H}_j$ and $B(\mathbf{c}_0, \epsilon)$ do not intersect, then $B(\mathbf{c}_0, \epsilon)$ completely belongs to $\mathcal{H}_j^+$ or $\mathcal{H}_j^-$. Without loss of generality, we can remove all $j$ such that $\mathcal{H}_j$ does not intersect $B(\mathbf{c}_0, \epsilon)$ and assume there are $k$ affine subspaces of $\mathbb{R}^n$ with dimension $n-1$ intersecting $B(\mathbf{c}_0, \epsilon)$. Let us sequentially carry out the intersection in (74). Every time before the operation of the $j$-th intersection between $B(\mathbf{c}_{j-1}, \frac{\epsilon}{2^{j-1}})$ and $\left( \mathcal{H}_j^+ \bigcup \mathcal{H}_j^- \right)$, there exists an open $\frac{\epsilon}{2^j}$-radius ball $B(\mathbf{c}_j, \frac{\epsilon}{2^j})$ for some $\mathbf{c}_j \in B(\mathbf{c}_{j-1}, \frac{\epsilon}{2^{j-1}})$ such that it does not intersect with $\mathcal{H}_j$. Therefore, at the end of the sequential process, there exists an open ball that does not intersect any of these $k$ affine subspaces of $\mathbb{R}^n$ with dimension $n-1$. The set in (74) is nonempty, implying (26) holds true. $\square$

### B.22 Proof of Proposition 1

*Proof.* We prove the claim by contraposition. If the number of closed connected subsets is not a minimum, i.e., Assumption 1 is not satisfied, then such a number can be decreased by merging some of the intersecting closed connected subsets that have the same corresponding affine functions. Therefore, there exist at least two closed connected subsets that can be made larger.

On the other hand, if the closed connected subsets, say $\mathcal{X}_1, \mathcal{X}_2, \cdots, \mathcal{X}_I$, have at least one of the subsets that can be made larger, then there exist at least two intersecting closed connected subsets, say $\mathcal{X}_i$ and $\mathcal{X}_j$, from $\mathcal{X}_1, \mathcal{X}_2, \cdots, \mathcal{X}_I$ such that their corresponding affine functions are the same. Otherwise, any closed connected subset cannot be made larger than itself. Therefore, $\mathcal{X}_i$ and $\mathcal{X}_j$ can be replaced with $\mathcal{X}_i \bigcup \mathcal{X}_j$ and these $I-1$ closed connected subsets still satisfy Definition 1, implying that $I$ is not the minimum. $\square$

## C Algorithms and time complexities

Table 1: The running time of Algorithm 1 is upper bounded by $\mathrm{poly}(n, k, q, L)$.

| Line | Operation count | Explanation |
|------|-----------------|-------------|
| 1 | $\mathcal{O}\left(nq \max(n^2, q)\right)$ | Algorithm 6 (see Table 6). |
| 2 | $\mathcal{O}(q)$ | Repeat Line 3 to Line 9 $q$ times. |
| 3 | $\mathcal{O}(1)$ | Initialize an empty placeholder. |
| 4 | $\mathcal{O}(k)$ | Repeat Line 5 to Line 7 $k$ times. |
| 5 | $\mathrm{poly}\,(n, q, L)$ | Solve a linear program [Vavasis and Ye, 1996]. |
| 6 | $\mathcal{O}(1)$ | Add an index. |
| 7 | - | - |
| 8 | - | - |
| 9 | $\mathcal{O}\left(k^2 \max(k \log_2 k, n)\right)$ | Algorithm 2 (see Table 2). |
| 10 | - | - |
| 11 | $\mathcal{O}\left(q \max(n, k)^2 \max(n, k, q) \log_2 k\right)$ | Algorithm 3 (see Table 3). |
| 12 | $\mathcal{O}\left(q^3 \log_2 q\right)$ | Algorithm 2 (see Table 2). |
| 13 | $\mathcal{O}\left(q^3 \max(n, k)^3 \log_2 q\right)$ | Algorithm 4 (see Table 4). |

**Algorithm 2** Find a ReLU network that computes the extremum of affine functions

---

**Input:** Scalar-valued affine functions $f_1, f_2, \cdots, f_m$ on $\mathbb{R}^n$ and the type of extremum (max or min).
**Output:** Parameters of an $l$-layer ReLU network $g$ computing $g(\mathbf{x}) = \max_{i \in [m]} .f_i(\mathbf{x})$ or $g(\mathbf{x}) = \min_{i \in [m]} .f_i(\mathbf{x})$ for all $\mathbf{x} \in \mathbb{R}^n$.

1: $\mathbf{A} \leftarrow \begin{bmatrix} -1 & 1 \\ 1 & 0 \\ -1 & 0 \end{bmatrix}, \mathbf{B} \leftarrow \begin{bmatrix} 1 & 1 & -1 \end{bmatrix}, \mathbf{C} \leftarrow \begin{bmatrix} 1 \\ -1 \end{bmatrix}$      ▷ Constant matrices

2: $\boldsymbol{\Psi}(\mathbf{Y}, \mathbf{Z}) \leftarrow \begin{bmatrix} \mathbf{Y} & \mathbf{0} \\ \mathbf{0} & \mathbf{Z} \end{bmatrix}$ ▷ A function generating a block diagonal matrix composed of $\mathbf{Y}$ and $\mathbf{Z}$

3: $\boldsymbol{\Phi}(\mathbf{Y}, s) \leftarrow \begin{bmatrix} \mathbf{Y}^{(1)} & \mathbf{0} & \cdots & \mathbf{0} \\ \mathbf{0} & \mathbf{Y}^{(2)} & \cdots & \mathbf{0} \\ \vdots & \vdots & \ddots & \vdots \\ \mathbf{0} & \mathbf{0} & \cdots & \mathbf{Y}^{(s)} \end{bmatrix}$ ▷ A block diagonal matrix with $\mathbf{Y}$ repeated $s$ times

4: $l \leftarrow \lceil \log_2 m \rceil + 1, k_0 \leftarrow n, k_l \leftarrow 1, c_0 \leftarrow m$     ▷ $l$ is the number of layers of $g$
5: **for** $i = 1, 2, \cdots, l - 1$ **do**
6:    **if** $c_{i-1}$ is even **then**
7:      $c_i \leftarrow \frac{c_{i-1}}{2}$
8:      $k_i \leftarrow 3c_i$           ▷ Output dimension of the $i$-th layer
9:    **else**
10:      $c_i \leftarrow \frac{c_{i-1}+1}{2}$
11:      $k_i \leftarrow 3c_i - 1$         ▷ Output dimension of the $i$-th layer
12:    **end if**
13: **end for**
14: $\mathbf{W}_1 \leftarrow \begin{bmatrix} \nabla f_1 & \nabla f_2 & \cdots & \nabla f_m \end{bmatrix}^{\mathsf{T}}, \mathbf{b}_1 \leftarrow \begin{bmatrix} f_1(0) & f_2(0) & \cdots & f_m(0) \end{bmatrix}^{\mathsf{T}}$
15: **if** $l > 1$ **then**          ▷ Find the weights of input and output layers, if any
16:    **if** $c_0$ is even **then**
17:      $\mathbf{W}_1 \leftarrow \boldsymbol{\Phi}\left(\mathbf{A}, c_1\right) \mathbf{W}_1, \mathbf{b}_1 \leftarrow \boldsymbol{\Phi}\left(\mathbf{A}, c_1\right) \mathbf{b}_1$
18:    **else**
19:      $\mathbf{W}_1 \leftarrow \boldsymbol{\Psi}\left(\boldsymbol{\Phi}\left(\mathbf{A}, c_1 - 1\right), \mathbf{C}\right) \mathbf{W}_1, \mathbf{b}_1 \leftarrow \boldsymbol{\Psi}\left(\boldsymbol{\Phi}\left(\mathbf{A}, c_1 - 1\right), \mathbf{C}\right) \mathbf{b}_1$
20:    **end if**
21:    $\mathbf{W}_l \leftarrow \mathbf{B}, \mathbf{b}_l \leftarrow \mathbf{0}_{k_l}$
22: **end if**
23: **if** $l > 2$ **then**           ▷ Find the weights of remaining layers, if any
24:    **for** $i = 2, 3, \cdots, l - 1$ **do**
25:      **if** $c_{i-1}$ is even **then**
26:        $\mathbf{T} \leftarrow \boldsymbol{\Phi}\left(\mathbf{A}, c_i\right)$
27:      **else**
28:        $\mathbf{T} \leftarrow \boldsymbol{\Psi}\left(\boldsymbol{\Phi}\left(\mathbf{A}, c_i - 1\right), \mathbf{C}\right)$
29:      **end if**
30:      **if** $c_{i-2}$ is even **then**
31:        $\mathbf{W}_i \leftarrow \mathbf{T}\boldsymbol{\Phi}\left(\mathbf{B}, c_{i-1}\right)$
32:      **else**
33:        $\mathbf{W}_i \leftarrow \mathbf{T}\boldsymbol{\Psi}\left(\boldsymbol{\Phi}\left(\mathbf{B}, c_{i-1} - 1\right), \mathbf{C}^{\mathsf{T}}\right)$
34:      **end if**
35:      $\mathbf{b}_i \leftarrow \mathbf{0}_{k_i}$
36:    **end for**
37: **end if**
38: **if** type of extremum is the minimum **then**
39:    $\mathbf{W}_1 \leftarrow -\mathbf{W}_1, \mathbf{b}_1 \leftarrow -\mathbf{b}_1$
40:    $\mathbf{W}_l \leftarrow -\mathbf{W}_l, \mathbf{b}_l \leftarrow -\mathbf{b}_l$
41: **end if**          ▷ See Table 2 in Appendix C for complexity analysis

---

---

**Algorithm 3** Find a ReLU network that concatenates a number of given ReLU networks

---

**Input:** Weights of $k$ ReLU networks $g_1, g_2, \cdots, g_k$ denoted by $\{\mathbf{W}_i^j, \mathbf{b}_i^j\}_{i=1}^{l_j}$ for $j \in [k]$.

**Output:** Parameters of an $l$-layer ReLU network $g$ computing $g(\mathbf{x}) = \begin{bmatrix} g_1(\mathbf{x}) \\ g_2(\mathbf{x}) \\ \vdots \\ g_k(\mathbf{x}) \end{bmatrix}, \forall \mathbf{x} \in \mathbb{R}^n$.

1: $l \leftarrow \max_{j \in [k]} l_j$

2: $\mathbf{W}_1 \leftarrow \begin{bmatrix} \mathbf{W}_1^1 \\ \mathbf{W}_1^2 \\ \vdots \\ \mathbf{W}_1^k \end{bmatrix}, \mathbf{b}_1 \leftarrow \begin{bmatrix} \mathbf{b}_1^1 \\ \mathbf{b}_1^2 \\ \vdots \\ \mathbf{b}_1^k \end{bmatrix}$ $\quad\quad\quad\quad\quad\quad\quad\quad$ ▷ Weights of the input layer

3: **for** $j = 1, 2, \cdots, k$ **do**
4: $\quad$ **if** $l_j < l$ **then** $\quad\quad\quad$ ▷ Append an identity mapping network to the network if it is shallower
5: $\quad\quad$ $m \leftarrow$ output dimsion of $g_j$
6: $\quad\quad$ $g_j^c \leftarrow$ run Algorithm 5 with an input dimension $m$ and a number of layers $l - l_j + 1$
7: $\quad\quad$ $g_j' \leftarrow$ run Algorithm 4 with $g_j$ and $g_j^c$
8: $\quad\quad$ $\{\mathbf{W}_i^j, \mathbf{b}_i^j\}_{i=1}^l \leftarrow$ weights of $g_j'$
9: $\quad$ **end if**
10: **end for**
11: **for** $i = 2, 3, \cdots, l$ **do** $\quad\quad\quad\quad\quad\quad\quad\quad\quad\quad\quad\quad$ ▷ Find the remaining weights

12: $\quad$ $\mathbf{W}_i \leftarrow \begin{bmatrix} \mathbf{W}_i^1 & \mathbf{0} & \cdots & \mathbf{0} \\ \mathbf{0} & \mathbf{W}_i^2 & \cdots & \mathbf{0} \\ \vdots & \vdots & \ddots & \vdots \\ \mathbf{0} & \mathbf{0} & \cdots & \mathbf{W}_i^k \end{bmatrix}, \mathbf{b}_i \leftarrow \begin{bmatrix} \mathbf{b}_i^1 \\ \mathbf{b}_i^2 \\ \vdots \\ \mathbf{b}_i^k \end{bmatrix}$

13: **end for** $\quad\quad\quad\quad\quad\quad\quad\quad\quad\quad$ ▷ See Table 3 in Appendix C for complexity analysis

---

---

**Algorithm 4** Find a ReLU network computing a composition of two given ReLU networks

---

**Input:** Weights of two ReLU networks $g_1$ and $g_2$ denoted by $\{\mathbf{W}_i^1, \mathbf{b}_i^1\}_{i=1}^{l_1}$ and $\{\mathbf{W}_i^2, \mathbf{b}_i^2\}_{i=1}^{l_2}$.
**Output:** Parameters of an $l$-layer ReLU network $g$ computing $g(\mathbf{x}) = g_2\big(g_1(\mathbf{x})\big), \forall \mathbf{x} \in \mathbb{R}^n$.
1: $l \leftarrow l_1 + l_2 - 1$
2: **for** $i = 1, 2, \cdots, l$ **do**
3: $\quad$ **if** $i < l_1$ **then** $\quad\quad\quad$ ▷ The first $l_1 - 1$ layers are identical to the corresponding layers in $g_1$
4: $\quad\quad$ $\mathbf{W}_i \leftarrow \mathbf{W}_i^1, \mathbf{b}_i \leftarrow \mathbf{b}_i^1$
5: $\quad$ **else if** $i = l_1$ **then** $\quad\quad\quad\quad$ ▷ A composition of affine functions is still an affine function
6: $\quad\quad$ $\mathbf{W}_i \leftarrow \mathbf{W}_1^2 \mathbf{W}_{l_1}^1, \mathbf{b}_i \leftarrow \mathbf{W}_1^2 \mathbf{b}_{l_1}^1 + \mathbf{b}_1^2$
7: $\quad$ **else** $\quad\quad\quad\quad\quad$ ▷ The last $l_2 - 1$ layers are identical to the corresponding layers in $g_2$
8: $\quad\quad$ $\mathbf{W}_i \leftarrow \mathbf{W}_{i-l_1+1}^2, \mathbf{b}_i \leftarrow \mathbf{b}_{i-l_1+1}^2$
9: $\quad$ **end if**
10: **end for** $\quad\quad\quad\quad\quad\quad\quad\quad\quad\quad\quad$ ▷ See Table 4 in Appendix C for complexity analysis

---

---

**Algorithm 5** Find a ReLU network that computes an identity mapping for a given depth

---

**Input:** The input dimension $n$ and the number of layers $l$ of the target ReLU network.
**Output:** Parameters of an $l$-layer ReLU network $g$ computing $g(\mathbf{x}) = \mathbf{x}, \forall \mathbf{x} \in \mathbb{R}^n$.

1: $\mathbf{A} \leftarrow \begin{bmatrix} 1 \\ -1 \end{bmatrix}, \mathbf{B} \leftarrow \begin{bmatrix} 1 & -1 \end{bmatrix}, \mathbf{C} \leftarrow \begin{bmatrix} 1 & -1 \\ -1 & 1 \end{bmatrix}$          $\triangleright$ Constant matrices

2: $\mathbf{\Phi}(\mathbf{Y}, s) = \begin{bmatrix} \mathbf{Y}^{(1)} & \mathbf{0} & \cdots & \mathbf{0} \\ \mathbf{0} & \mathbf{Y}^{(2)} & \cdots & \mathbf{0} \\ \vdots & \vdots & \ddots & \vdots \\ \mathbf{0} & \mathbf{0} & \cdots & \mathbf{Y}^{(s)} \end{bmatrix}$      $\triangleright$ A block diagonal matrix with $\mathbf{Y}$ repeated $s$ times

3: $k_0 \leftarrow n, k_l \leftarrow n, \mathbf{b}_l \leftarrow \mathbf{0}_n$
4: **for** $i = 1, 2, \cdots, l-1$ **do**
5:      $k_i \leftarrow 2n$          $\triangleright$ The number of hidden neurons at the $i$-th hidden layer
6:      $\mathbf{b}_i \leftarrow \mathbf{0}_{k_i}$
7: **end for**
8: **if** $l = 1$ **then**          $\triangleright$ Find the weights of input and output layers, if any
9:      $\mathbf{W}_1 \leftarrow \mathbf{I}_{n \times n}$          $\triangleright$ An identity matrix
10: **else**
11:      $\mathbf{W}_1 \leftarrow \mathbf{\Phi}(\mathbf{A}, k_0)$
12:      $\mathbf{W}_l \leftarrow \mathbf{\Phi}(\mathbf{B}, k_l)$
13: **end if**
14: **if** $l > 2$ **then**          $\triangleright$ Find the weights of hidden layers, if any
15:      **for** $i = 2, 3, \cdots, l-1$ **do**
16:          $\mathbf{W}_i \leftarrow \mathbf{\Phi}(\mathbf{C}, n)$
17:      **end for**
18: **end if**          $\triangleright$ See Table 5 in Appendix C for complexity analysis

---

**Algorithm 6** Find all distinct linear components of a CPWL function

---

**Input:** An unknown CPWL function $p$ whose output can be observed by feeding input from $\mathbb{R}^n$ to the function. A center $\mathbf{c}_i$ and radius $\epsilon_i > 0$ of any closed $\epsilon_i$-radius ball $B(\mathbf{c}_i, \epsilon_i)$ such that $B(\mathbf{c}_i, \epsilon_i) \subset \mathcal{X}_i$ for $i = 1, 2, \cdots, q$ where $\{\mathcal{X}_i\}_{i \in [q]}$ are all pieces of $p$.
**Output:** All distinct linear components of $p$, denoted by $\mathcal{F}$.

1: $\mathcal{F} \leftarrow \emptyset$          $\triangleright$ Initialize the set of all distinct linear components
2: **for** $i = 1, 2, \cdots, q$ **do**
3:      $\mathbf{x}_0 \leftarrow \mathbf{c}_i$          $\triangleright$ select the center of $B(\mathbf{c}_i, \epsilon_i)$
4:      $y_0 \leftarrow p(\mathbf{x}_0)$
5:      $\begin{bmatrix} \mathbf{s}_1 & \mathbf{s}_2 & \cdots & \mathbf{s}_n \end{bmatrix} \leftarrow \epsilon_i \mathbf{I}_{n \times n}$          $\triangleright$ scale the standard basis of $\mathbb{R}^n$
6:      $\mathbf{S} \leftarrow \begin{bmatrix} \mathbf{s}_1 & \mathbf{s}_2 & \cdots & \mathbf{s}_n \end{bmatrix}$
7:      $\mathbf{z} \leftarrow \begin{bmatrix} p(\mathbf{s}_1 + \mathbf{x}_0) - y_0 \\ p(\mathbf{s}_2 + \mathbf{x}_0) - y_0 \\ \vdots \\ p(\mathbf{s}_n + \mathbf{x}_0) - y_0 \end{bmatrix}$
8:      $\mathbf{a} \leftarrow \mathbf{S}^{-\top} \mathbf{z}$          $\triangleright$ Find the linear map by solving a system of linear equations
9:      $b \leftarrow y_0 - \mathbf{a}^\top \mathbf{x}_0$          $\triangleright$ Find the translation
10:      $f \leftarrow \mathbf{x} \mapsto \mathbf{a}^\top \mathbf{x} + b$          $\triangleright$ The affine map on $\mathcal{X}_i$
11:      **if** $f \notin \mathcal{F}$ **then**    $\triangleright$ Only add the affine map $f$ to the set $\mathcal{F}$ if $f$ is distinct to all elements of $\mathcal{F}$
12:          $\mathcal{F} \leftarrow \mathcal{F} \bigcup \{f\}$
13:      **end if**
14: **end for**          $\triangleright$ See Table 6 in Appendix C for complexity analysis

---

Table 2: The time complexity of Algorithm 2 is $\mathcal{O}\left(m^2 \max(m \log_2 m, n)\right)$.

| Line | Operation count | Explanation |
|---|---|---|
| 1 | $\mathcal{O}(1)$ | Initialize constant matrices. |
| 2 | $\mathcal{O}\left(d_1^2\right)$ | Let $d_1$ be the maximum dimension of $\mathbf{Y}$ and $\mathbf{Z}$. |
| 3 | $\mathcal{O}(s^2 d_2^2)$ | Let $d_2$ be the maximum dimension of $\mathbf{Y}$. |
| 4 | $\mathcal{O}(1)$ | Scalar assignments. |
| 5 | $\mathcal{O}(\log_2 m)$ | Repeat Line 6 to Line 12 $\lceil \log_2 m \rceil$ times. |
| 6 | $\mathcal{O}(1)$ | Check a scalar is even or not. |
| 7 | $\mathcal{O}(1)$ | Compute a scalar. |
| 8 | $\mathcal{O}(1)$ | Compute a scalar. |
| 9 | - | - |
| 10 | $\mathcal{O}(1)$ | Compute a scalar. |
| 11 | $\mathcal{O}(1)$ | Compute a scalar. |
| 12 | - | - |
| 13 | - | - |
| 14 | $\mathcal{O}(mn)$ | Assign a matrix and a vector. |
| 15 | $\mathcal{O}(1)$ | Check a scalar inequality. |
| 16 | $\mathcal{O}(1)$ | Check a scalar is even or not. |
| 17 | $\mathcal{O}(m^2 n)$ | Matrix creation and multiplication. |
| 18 | - | - |
| 19 | $\mathcal{O}(m^2 n)$ | Matrix creation and multiplication. |
| 20 | - | - |
| 21 | $\mathcal{O}(1)$ | Assign a constant matrix and vector. |
| 22 | - | - |
| 23 | $\mathcal{O}(1)$ | Check a scalar inequality. |
| 24 | $\mathcal{O}(\log_2 m)$ | Repeat Line 25 to Line 30 $\lceil \log_2 m \rceil - 1$ times. |
| 25 | $\mathcal{O}(1)$ | Check a scalar is even or not. |
| 26 | $\mathcal{O}(m^2)$ | Matrix creation. |
| 27 | - | - |
| 28 | $\mathcal{O}(m^2)$ | Matrix creation. |
| 29 | - | - |
| 30 | $\mathcal{O}(1)$ | Check a scalar is even or not. |
| 31 | $\mathcal{O}(m^3)$ | Matrix creation and multiplication. |
| 32 | - | - |
| 33 | $\mathcal{O}(m^3)$ | Matrix creation and multiplication. |
| 34 | - | - |
| 35 | $\mathcal{O}(m)$ | Assign a vector whose length is at most $\left\lceil \frac{3m}{2} \right\rceil$. |
| 36 | - | - |
| 37 | - | - |
| 38 | $\mathcal{O}(1)$ | Check the binary data type. |
| 39 | $\mathcal{O}(mn)$ | Reverse the sign of $\mathbf{W}_1$ and $\mathbf{b}_1$. |
| 40 | $\mathcal{O}(1)$ | Reverse the sign of a constant matrix and a constant bias. |
| 41 | - | - |

Table 3: The time complexity of Algorithm 3 is $\mathcal{O}\left(d^2 kl \max(d, k)\right)$ where $d$ is the maximum dimension of all the weight matrices in $g_1, g_2, \cdots, g_k$ and $l = \max_{j \in [k]} l_j$.

| Line | Operation count | Explanation |
|------|-----------------|-------------|
| 1 | $\mathcal{O}(k)$ | Find the maximum among $k$ numbers. |
| 2 | $\mathcal{O}(d^2 k)$ | Matrix concatenation and assignment. |
| 3 | $\mathcal{O}(k)$ | Repeat Line 4 to Line 9 $k$ times. |
| 4 | $\mathcal{O}(1)$ | Check a scalar inequality. |
| 5 | $\mathcal{O}(1)$ | A scalar assignment. |
| 6 | $\mathcal{O}(d^2 l)$ | Algorithm 5 (see Table 5). |
| 7 | $\mathcal{O}(d^3 l)$ | Algorithm 4 (see Table 4). |
| 8 | $\mathcal{O}(d^2 l)$ | Assign weights of the network. |
| 9 | - | - |
| 10 | - | - |
| 11 | $\mathcal{O}(l)$ | Repeat Line 12 $l - 1$ times. |
| 12 | $\mathcal{O}(d^2 k^2)$ | Assign a matrix and a vector. |
| 13 | - | - |

Table 4: The time complexity of Algorithm 4 is $\mathcal{O}\left(d^3 \max(l_1, l_2)\right)$ where $d$ is the maximum dimension of all the weight matrices in $g_1$ and $g_2$.

| Line | Operation count | Explanation |
|------|-----------------|-------------|
| 1 | $\mathcal{O}(1)$ | Assign a constant. |
| 2 | $\mathcal{O}(l)$ | Repeat Line 3 to Line 9 $l$ times. |
| 3 | $\mathcal{O}(1)$ | Check a scalar inequality. |
| 4 | $\mathcal{O}(d^2)$ | Assign a matrix and a vector (at most $d^2 + d$ elements). |
| 5 | $\mathcal{O}(1)$ | Check a scalar equality. |
| 6 | $\mathcal{O}(d^3)$ | Matrix multiplication and assignment. |
| 7 | - | - |
| 8 | $\mathcal{O}(d^2)$ | Assign a matrix and a vector (at most $d^2 + d$ elements). |
| 9 | - | - |
| 10 | - | - |

Table 5: The time complexity of Algorithm 5 is $\mathcal{O}(n^2 l)$.

| Line | Operation count | Explanation |
|------|-----------------|-------------|
| 1 | $\mathcal{O}(1)$ | Initialize constant matrices. |
| 2 | $\mathcal{O}(s^2 d_1 d_2)$ | Create a block diagonal matrix from $\mathbf{Y} \in \mathbb{R}^{d_1 \times d_2}$ and $s \in \mathbb{N}$. |
| 3 | $\mathcal{O}(n)$ | Assign two constant scalars and one constant vector of length $n$. |
| 4 | $\mathcal{O}(l)$ | Repeat Line 5 to line 6 $l$ times. |
| 5 | $\mathcal{O}(1)$ | Assign a scalar. |
| 6 | $\mathcal{O}(n)$ | Assign a vector whose length $k_i$ is equal to $2n$. |
| 7 | - | - |
| 8 | $\mathcal{O}(1)$ | Check a scalar equality. |
| 9 | $\mathcal{O}(n^2)$ | Assign an $n$-by-$n$ matrix. |
| 10 | - | - |
| 11 | $\mathcal{O}(n^2)$ | Assign a $2n$-by-$n$ block diagonal matrix. |
| 12 | $\mathcal{O}(n^2)$ | Assign an $n$-by-$2n$ block diagonal matrix. |
| 13 | - | - |
| 14 | $\mathcal{O}(1)$ | Check a scalar inequality. |
| 15 | $\mathcal{O}(l)$ | Repeat Line 16 $l - 2$ times. |
| 16 | $\mathcal{O}(n^2)$ | Assign a $2n$-by-$2n$ block diagonal matrix. |
| 17 | - | - |
| 18 | - | - |

Table 6: The time complexity of Algorithm 6 is $\mathcal{O}\left(nq\max(n^2, q)\right)$.

| Line | Operation count | Explanation |
|------|-----------------|-------------|
| 1 | $\mathcal{O}(1)$ | Initialize an empty placeholder $\mathcal{F}$. |
| 2 | $\mathcal{O}(q)$ | Repeat Line 3 to line 13 $q$ times. |
| 3 | $\mathcal{O}(1)$ | Select an interior point. Use the center of the ball. |
| 4 | $\mathcal{O}(1)$ | Evaluate the function on the point. |
| 5 | $\mathcal{O}(n^2)$ | Scale and assign an $n$-by-$n$ matrix. |
| 6 | - | - |
| 7 | $\mathcal{O}(n)$ | Translate, evaluate, and subtract $n$ points. |
| 8 | $\mathcal{O}(n^3)$ | Solve a system of $n$ linear equations with $n$ variables. |
| 9 | $\mathcal{O}(n)$ | Solve the translation term in the affine map |
| 10 | - | - |
| 11 | $\mathcal{O}(nq)$ | Each affine map has $n+1$ parameters and $\mathcal{F}$ has at most $q$ elements. |
| 12 | $\mathcal{O}(1)$ | Add a distinct affine map to $\mathcal{F}$. |
| 13 | - | - |
| 14 | - | - |

# D  Open source implementation and run time of Algorithm 1

We implement Algorithm 1 in Python. Figure 3 shows that the run time of the algorithm is greatly affected by the number of pieces $q$.

Code is available at https://github.com/kjason/CPWL2ReLUNetwork.

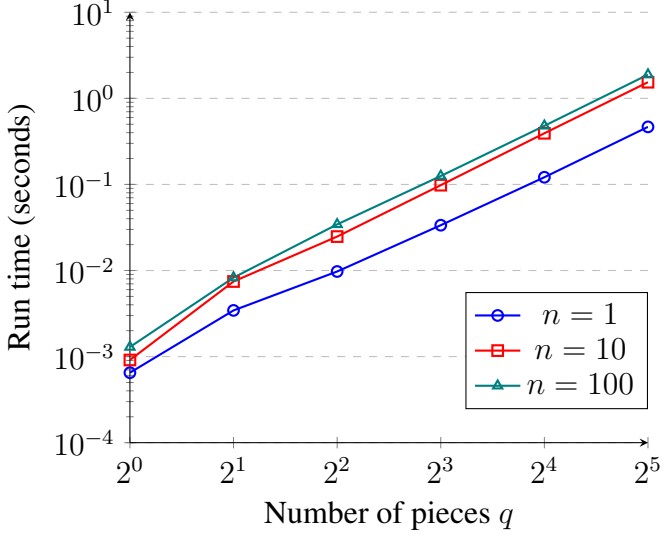

Figure 3: The run time of Algorithm 1 is an average of 50 trials. Every trial runs Algorithm 1 with a random CPWL function whose input dimension is $n$ and number of pieces is $q$. The code provided in the above link is run on a computer (Microsoft Surface Laptop Studio) with the Intel Core i7-11370H.