# OpenReview forum: "Improved Bounds on Neural Complexity for Representing Piecewise Linear Functions"
_NeurIPS.cc/2022/Conference — NeurIPS 2022 Accept_

### Official Review · Reviewer_snmk · 2022-07-11

**Rating:** 6
**Confidence:** 4
**Soundness:** 4 excellent
**Presentation:** 3 good
**Contribution:** 3 good

**Summary:**

Summary. The authors study the representation of continuous piecewise linear (CPWL) functions using neural networks with ReLU activations. They showed that, for Any CPWL function with q pieces or k linear components, it can be represented by a ReLU network
with certain layers l, maximum width w, and the number of hidden neurons h. The bounds in this paper are much better than in previous papers. The algorithm for constructing such networks is also given.

**Questions:**

It would be interesting to extend the results to more general network architectures.

**Limitations:**

Yes, the authors have adequately addressed the limitations and potential negative societal impact of their work.

**Strengths And Weaknesses:**

Originality: The related works are adequately cited. The novelty of this paper is high. The results on the sharp bound of number of hidden neurons needed to representing CPWL functions in this paper, will certainly help us have a better understating of representation power of neural networks from a theoretical way. I have checked the technique parts and find that the proofs are solid. I think this is a significant contribution to the machine learning immunity.

Quality: This paper is technically sound.

Clarity: This paper is clearly written and well organized. I find it easy to follow.

Significance: I think the results in this paper is significant, as explained above.

---

> ### Author Response · Authors · 2022-08-02
> **Extending the results to more general network architectures**
>
> First, we would like to express our gratitude to the reviewer for checking the proofs. Thank you very much for reviewing the paper and recognizing the significant contribution of the work.
>
> ### Below we respond to the reviewer’s suggestion:
> >It would be interesting to extend the results to more general network architectures.
>
> **High-level response:**
>
> We are encouraged by the reviewer’s suggestion. Depending on the architecture and the activation function, one may be able to derive some bounds that are better than the quadratic complexity. However, for architectures that impose more constraints such as residual networks, it is still not clear how we can derive better bounds for them.
>
> **Details and insights:**
>
> Given a network architecture, one would be interested in finding a bound for the number of hidden neurons required to represent any given CPWL function. To shed more light on this, we take residual networks as an illustrating example and point out the hope and difficulty to obtain a better bound for the number of hidden neurons. Because a different architecture may rely on a very different technique, we limit the discussion on residual networks (ResNets) here. The network architecture we consider is defined by the following statement.
>
> Let $l$ be an even positive integer. A function $g:\mathbb{R}^{k_0}\to\mathbb{R}^{k_{l}}$ is an $l$-layer ResNet if there exist weights $\mathbf{W}\_i\in\mathbb{R}^{k\_{i}\times k\_{i-1}}$ and $\mathbf{b}\_i\in\mathbb{R}^{k\_i}$ for $i\in[l]$ such that the input-output relationship of the network satisfies
>     $
>         g(\mathbf{x})=h_l(\mathbf{x})
>     $
>     where $h_1(\mathbf{x})=\mathbf{W}\_1\mathbf{x}+\mathbf{b}\_1$, $h\_l(\mathbf{x})=\mathbf{W}\_{l}h\_{l-1}(\mathbf{x})+\mathbf{b}\_{l}$, and
>     \begin{equation}
>         h\_i(\mathbf{x})=\mathbf{W}\_{i}\sigma\_{k\_{i-1}}\left(h\_{i-1}(\mathbf{x})\right)+\mathbf{b}\_{i}
>     \end{equation}
> if $i$ is even, and
> \begin{equation}
> h\_i(\mathbf{x})=h\_{i-2}(\mathbf{x})+\mathbf{W}\_{i}\sigma\_{k_{i-1}}\left(h\_{i-1}(\mathbf{x})\right)+\mathbf{b}\_{i}
> \end{equation}
> if $i$ is odd for every $i\in[l]\setminus \\{1,l\\}$.
>
> **Hope:**
> Before finding a bound, we first recognize that finding a bound may be possible due to the argument that “Any ReLU ResNet represents a CPWL function.” One can prove this statement by using the results in the plain ReLU network and the fact that (a) the addition of two CPWL functions is still CPWL and (b) the composition of two CPWL functions is still CPWL.
>
> **Difficulty:**
> It is still not clear how we can find a bound that is better than the quadratic complexity. Because the skip connection for every residual block in a ResNet maintains the residual dimension in every block, the number of output neurons in the last layer of every block is the same. Such a constraint makes it difficult to fully utilize the max-min representation of CPWL functions in Eq. (12).
>
> ---
>
> Given the positive comments below:
> >“The novelty of this paper is high.”
>
> >“I think this is a significant contribution to the machine learning community."
>
> >“This paper is clearly written and well organized.”
>
>
> We are surprised by a rating of 6. Based on the comments from the other reviewers, we added new results including the complexity analysis and the relaxation of the assumption in Algorithm 1 by adding a new algorithm ( Algorithm 6). We now feel the contribution of the paper has been further enhanced. We sincerely hope the reviewer will reconsider the rating based on the responses to the issues raised and the new results provided.

---

> ### Author Response · Authors · 2022-08-08
> **We would be happy to take any further questions if any**
>
> We thank the reviewer for providing valuable comments that help us strengthen the paper. Given the time constraint of the discussion period, we would like to let the reviewer know that we would be happy to answer any further questions if any. Thank you very much again for taking the time and effort in reviewing the paper and checking the proofs.

---

### Official Review · Reviewer_WJTP · 2022-07-12

**Rating:** 7
**Confidence:** 3
**Soundness:** 3 good
**Presentation:** 3 good
**Contribution:** 3 good

**Summary:**

In this paper the authors provide bounds for approximating a continuous piecewise linear function by a ReLU function.

When compared to previous work, the bounds provided by the authors are tighter. Furthermore, the bounds provided by the authors only depend upon the total number of linear components in the CPWL (unlike previous work). This helps when the growth in the number of dimensions is faster than the number of linear components. The authors are able to do this because they make use of a different representation of a CPWL function (as mentioned in equation 13).

The authors also provide an algorithm for finding a ReLU network that computes the given a CPWL. However, this algorithm requires one to know how many linear components are present in the function beforehand, and uses the fact that the final function is a combination of smaller ReLU function approximating the linear components.

**Questions:**

I think giving a sketch of what Algorithms 2, 3 and 4 do in the main text—even if it is just a sketch will improve the readability of the paper.

**Limitations:**

Yes

**Strengths And Weaknesses:**

The paper is very well written and motivated and provides a rigorous proof for improving of bounds for approximating CPWL functions with neural network. The authors have done an amazing job at motivating and explaining their results as well as placing their work with the related work (which is very well explained). I really enjoyed reading the paper.

A possible drawback is that Algorithm 1 assumes that the number of linear components in the function is known beforehand. Is there a way that this assumption could be relaxed in a way?

---

> ### Author Response · Authors · 2022-08-02
> **We improved the generality of Algorithm 1**
>
> Thank you very much for reviewing the paper. We are glad to hear that the reviewer really enjoyed reading the paper.
>
> ### Below we respond to the reviewer’s suggestions:
> >However, this algorithm requires one to know how many linear components are present in the function beforehand
>
> >A possible drawback is that Algorithm 1 assumes that the number of linear components in the function is known beforehand. Is there a way that this assumption could be relaxed in a way?
>
> We thank the reviewer for this important comment. We agree with the reviewer that this is a drawback and we believe relaxing such an assumption greatly improves the generality of Algorithm 1. Reflecting on this comment, we worked on Algorithm 1 and found a way to relax such an assumption by using the result from Lemma 12(a) which guarantees that the interior of each of the pieces is nonempty. In light of this, we have added a new algorithm in our revised manuscript, Algorithm 6 in Appendix C in the supplementary material, to find all linear components for Algorithm 1. Algorithm 1 now does not need to know any linear components or be given the number of linear components beforehand. It only needs to be given the pieces of the CPWL function p and be able to observe the output of p when feeding an input. We also added an explanation for Algorithm 6 in the main text (see Line 252 to Line 257 in the revised manuscript). Algorithm 6 is given below:
>
> ---
> ### **Algorithm 6**
>
> **Input**: An unknown CPWL function $p$ whose output can be observed by feeding input from $\mathbb{R}^n$
> to the function. A center $c_i$ and radius $\epsilon_i > 0$ of any closed $\epsilon$-radius ball $B(\mathbf{c}_i, \epsilon_i)$ such that $B(\mathbf{c}_i, \epsilon_i)\subset\mathcal{X}_i$ for $i=1,2,\cdots,q$ where $\mathcal{X}_1,\mathcal{X}_2,\cdots,\mathcal{X}_q$ are all pieces of $p$.
>
> **Output**: All distinct linear components of $p$, denoted by $\mathcal{F}$.
>
> $\quad$$\ \ $1: $\mathcal{F}\gets\emptyset$
>
> $\quad$$\ \ $2: **For** $i=1,2,\cdots,q$ **do**
>
> $\quad$$\ \ $3: $\quad$$\mathbf{x}_0 \gets \mathbf{c}_i$
>
> $\quad$$\ \ $4: $\quad$$y_0 \gets p(\mathbf{x}_0)$
>
> $\quad$$\ \ $5: $\quad$$\begin{bmatrix}\mathbf{s}_1&\mathbf{s}_2&\cdots&\mathbf{s}_n\end{bmatrix} \gets \epsilon_i\mathbf{I}_n$
>
> $\quad$$\ \ $6: $\quad$$\mathbf{S} \gets \begin{bmatrix}\mathbf{s}_1&\mathbf{s}_2&\cdots&\mathbf{s}_n\end{bmatrix}$
>
> $\quad$$\ \ $7: $\quad$$\mathbf{z} \gets \begin{bmatrix}p(\mathbf{s}_1+\mathbf{x}_0)-y_0 & p(\mathbf{s}_2+\mathbf{x}_0)-y_0 & \cdots & p(\mathbf{s}_n+\mathbf{x}_0)-y_0\end{bmatrix}^\mathsf{T}$
>
> $\quad$$\ \ $8: $\quad$$\mathbf{a} \gets \mathbf{S}^{-\mathsf{T}}\mathbf{z}$
>
> $\quad$$\ \ $9: $\quad$$b \gets y_0-\mathbf{a}^{\mathsf{T}}\mathbf{x}_0$
>
> $\quad$10: $\quad$$f \gets \mathbf{x}\mapsto\mathbf{a}^{\mathsf{T}}\mathbf{x}+b$
>
> $\quad$11: $\quad$**if** $f\not\in\mathcal{F}$ **then**
>
> $\quad$12: $\quad$$\quad$$\mathcal{F} \gets \mathcal{F}\bigcup\{f\}$
>
> $\quad$13: $\quad$**end if**
>
> $\quad$14: **end for**
>
> ---
>
> >I think giving a sketch of what Algorithms 2, 3 and 4 do in the main text—even if it is just a sketch will improve the readability of the paper.
>
> Yes, we agree with the reviewer that a simple sketch for each algorithm is beneficial for the reader. Algorithm 2, 3, and 4 are originally described right before the introduction of Theorem 2, Line 253 to Line 257. As it might not be clear enough in our previous manuscript, we have moved their description to the beginning of the paragraph, Line 247 to Line 251, in the revised manuscript.
>
> ---
> Based on the new algorithm (Algorithm 6) and the relaxation of the assumption in Algorithm 1, now the contribution of the paper has been further enhanced. We sincerely hope the reviewer finds that the paper has been technically strengthened.

---

> > ### Comment · Reviewer_WJTP · 2022-08-09
> > **Thank you for the Reply**
> >
> > The authors have addressed all my concerns and I also appreciate the effort the authors have put in clarifying the algorithm under conditions when the number of linear components are unknown. Thank you!

---

> > > ### Author Response · Authors · 2022-08-09
> > > **Thank you**
> > >
> > > Thank you for letting us know! Again, we thank the reviewer for taking the time and effort to review the paper and provide thoughtful comments.

---

> ### Author Response · Authors · 2022-08-08
> **We would be happy to take any further questions if any**
>
> We thank the reviewer for providing insightful suggestions that help us make the paper technically stronger. Given the time constraint of the discussion period, we would like to let the reviewer know that we would be happy to answer any further questions if any. Thank you very much again for taking the time and effort in reviewing the paper.

---

### Official Review · Reviewer_LuVg · 2022-07-21

**Rating:** 6
**Confidence:** 2
**Soundness:** 3 good
**Presentation:** 3 good
**Contribution:** 3 good

**Summary:**

The paper focuses on neural networks with ReLU activations. For these networks, the paper proposes bounds on the number of neurons needed to represent continuous piecewise linear functions (CPWL). These bounds are derived in terms of the number of linear pieces and the number of distinct linear components of the CPWL. In contrast to prior work, which proposed exponential bounds, the paper shows that quadratic bounds in the number of pieces are enough.

**Questions:**

**For the rebuttal**
* Can Algorithm 1 be used in practice? What is the computational complexity of the algorithm?

**Questions**
* L11: Why is the invariance wrt the input dimension more important than the quadratic bounds?
* L100: “q is always not less than k” -> $k \leq q$


**Limitations:**

Theoretical work, no direct negative societal impact.

**Strengths And Weaknesses:**

### Strengths
* The paper significantly improves previous bounds, which enables much more accurate estimates of the number of neurons needed.

### Weaknesses
* It is unclear whether Algorithm 1 can be used in practice. Also, the run time and the computational complexity of the algorithm are unclear.
**Clarity**
* Definition 4: ReLU network should be called ReLU-Multi Layer Perceptron. A general ReLU network is a network with only ReLU activations. A ResNet can also be a ReLU network. Hence, in the Broader impact chapter, it’s confusing that the text says “We focus on ReLU networks in this paper, but it is possible to derive bounds with similar asymptotic growth rates for other neural network architectures such as … residual networks …, densely connected networks …

---

> ### Author Response · Authors · 2022-08-02
> **Algorithm 1 is a polynomial time algorithm and our responses**
>
> We thank the reviewer for reviewing the paper and providing valuable comments. We have revised the manuscript based on your comments.
>
> ### Below we address specific concerns and suggestions:
> >It is unclear whether Algorithm 1 can be used in practice… Can Algorithm 1 be used in practice?
>
> **Clarification of Algorithm 1**
>
> Upon reflecting on your comment and Reviewer WJTP's comment, we realized we can relax the information needed for implementing Algorithm 1.  Algorithm 1 only needs to be given a closed $\epsilon$-ball in the interior of every piece of a CPWL function $p$ and be able to observe the output of $p$ when feeding an input. Algorithm 1 does not explicitly need to know any linear components.
>
> **Practical applications**
> 1. **DNN pruning:** Theorem 1 says that the number of hidden neurons of a ReLU MLP representing a CPWL function $p$ can be bounded by a quadratic function of the number of pieces of $p$. Such a result can have an impact on DNN pruning. Taking ReLU MLP for example, one may want to estimate how many neurons can be possibly pruned away for an application. If the number of linear regions or pieces required is given, known, or estimated (based on loose bounds of activation patterns) for the application, then one can leverage the quadratic bound to know at least how many neurons can be pruned or even directly construct a network that has a lower complexity by Algorithm 1.
> 2. **Neural engines**: A compute-intensive application computing CPWL functions can potentially leverage specialized hardware like neural engines to make the computation energy-efficient, given that neural engines are highly optimized for neural networks. Our Algorithm 1 provides an important procedure to convert a CPWL function to ReLU MLP. Meanwhile, it also guarantees the computational resources are bounded by a quadratic function, which is a substantial improvement compared to previous exponential bounds.
>
> >...the run time and the computational complexity of the algorithm are unclear… What is the computational complexity of the algorithm?
>
> We have worked on the time complexity of Algorithm 1 and below we show that Algorithm 1 is a polynomial time algorithm, $\text{poly}(n,q)$ for Theorem 1 and $\text{poly}(n,k,q)$ for Theorem 2. In fact, the time complexity is $\mathcal{O}\left(q^3\max(n,k)^3\log_2q\right)$. Based on the new results, we have updated several statements in the manuscript to reflect the time complexity.
>
> ### Time complexity of Algorithm 1
> | Line | Operation count | Explanation |
> | :--- | :--- | :--- |
> | 1 | $\mathcal{O}(nq\max(n^2,q))$ | Algorithm 6 (see Table 6). |
> |2 | $\mathcal{O}(q)$ | Repeat Line 3 to Line 9 $q$ times. |
> |3 | $\mathcal{O}(1)$ | Initialize an empty placeholder.|
> |4 | $\mathcal{O}(k)$ | Repeat Line 5 to Line 7 $k$ times.|
> |5 | $\mathcal{O}(n)$ | Check $n+1$ affinely independent vectors.|
> |6 | $\mathcal{O}(1)$ | Add an index.|
> |7 | - | -|
> |8 | - | -|
> |9 |  $\mathcal{O}\left(k^2\max(k\log_2k,n)\right)$ | Algorithm 2 (see Table 2).|
> |10 | - | -|
> |11 | $\mathcal{O}\left(q\max(n,k)^2\max(n,k,q)\log_2k\right)$ | Algorithm 3 (see Table 3).|
> |12 | $\mathcal{O}\left(q^3\log_2q\right)$ | Algorithm 2 (see Table 2).|
> |13 | $\mathcal{O}\left(q^3\max(n,k)^3\log_2q\right)$ | Algorithm 4 (see Table 4).|
> ---
>
> >Definition 4: ReLU network should be called ReLU-Multi Layer Perceptron. A general ReLU network is a network with only ReLU activations. A ResNet can also be a ReLU network. Hence, in the Broader impact chapter, it’s confusing that...
>
> We thank the reviewer for pointing this out. We primarily follow the terminology used by [Arora et al., 2018] and [He et al., 2020] (see the references below). We have added a sentence before Definition 4 to emphasize that the ReLU network considered in the paper is a ReLU multi-layer perceptron. We have also improved the broader impact section by clarifying the architecture used in the paper. We believe now the reader can clearly distinguish our network architecture from other advanced architectures.
>
> [^1]: R. Arora, A. Basu, P. Mianjy, and A. Mukherjee. Understanding deep neural networks with rectified linear units. In International Conference on Learning Representations, 2018.
>
> [^2]: J. He, L. Li, J. Xu, and C. Zheng. ReLU deep neural networks and linear finite elements. Journal of Computational Mathematics, 38(3):502–527, 2020.
>
> >L11: Why is the invariance wrt the input dimension more important than the quadratic bounds?
>
> We strongly agree with the observation that one is not more important than the other and that in reality, they are not comparable. We have improved this by removing “more importantly.”
>
> >L100: “q is always not less than k” -> $k\leq q$
>
> We have adopted the reviewer’s suggestion and improved the sentence. Thank you!
>
> ---
>
> Based on the revisions, now the contribution of the paper has been further enhanced. We sincerely hope the reviewer will reconsider the rating based on the responses to the issues raised and the new results provided.

---

> > ### Comment · Reviewer_LuVg · 2022-08-08
> > **Reply to Rebuttal**
> >
> > I would like to thank the authors for their clarifications and for the time complexity of the algorithm.
> >
> > When I asked whether Algorithm 1 can be used in practice, I was actually not so much looking for possible applications, but I was more interested on whether the algorithm can be implemented and run for standard networks on a reasonably fast computer. E.g., if you take some arbitrary 100-dimensional CPWL with 1 million pieces, how long would the algorithm take to run on a computer?
> > (It may not be possible to provide an answer to this question until the rebuttal deadline and this is not a problem for me. Maybe this information and an open source implementation of the algorithm could be added in a camera ready version. But since the paper's main application are the improved bounds and not the algorithm itself, this is not absolutely necessary.)
> >
> > I am happy with the authors' replies and recommend the acceptance of the paper.

---

> > > ### Author Response · Authors · 2022-08-09
> > > **We will measure the run time of the algorithm on a computer and add their results when possible**
> > >
> > > Thank you very much for replying to our rebuttal. We are glad to know that the reviewer is happy with our responses. We will add an open source implementation of Algorithm 1 and measure its run time on a computer under different conditions such as different numbers of pieces and input dimensions. These extra materials will be added to the supplementary material in the camera-ready version when possible. We highly appreciate the reviewer’s clarification and the concrete example provided. Finally, we would like to express our gratitude to the reviewer for recommending the acceptance of the paper.

---

> > > ### Author Response · Authors · 2022-08-09
> > > **Rating**
> > >
> > > If you see it fit, please consider giving the paper a higher rating to ensure acceptance. Thank you!

---

> ### Author Response · Authors · 2022-08-08
> **We would be happy to take any further questions if any**
>
> We thank the reviewer for providing valuable suggestions that help us make the paper technically stronger. Given the time constraint of the discussion period, we would like to let the reviewer know that we would be happy to answer any further questions if any. Thank you very much again for taking the time and effort in reviewing the paper.

---

### Meta-Review · Area_Chair_Vv3p · 2022-08-27

**Recommendation:** Accept
**Confidence:** Certain

**Metareview:**

Three reviewers agree that this work meets the bar for acceptance, rating it weak accept, weak accept, and accept. The work provides bounds for approximating continuous piecewise linear functions by ReLU networks and an algorithm. Reviewers praised the novelty and significance, and were positive about clarifications offered during the discussion period, particularly about the time complexity of the algorithm. Hence I am recommending accept. I encourage the authors to still work on the items of the discussion and the promised additions such as the open source implementation of their algorithm for the final version of the manuscript.

**Award:**

No

---

### Decision · Program_Chairs · 2022-09-14

Accept